# CPQS-Tuning: A Model Self-Perception-Based Data Filtering Algorithm for Efficient Instruction Fine-Tuning

**Yi Ren**
Key Laboratory for Novel Software Technology
Nanjing University
Nanjing, China
`602025320012@smail.nju.edu.cn`

**Yanhui Li**[*]
Key Laboratory for Novel Software Technology
Nanjing University
Nanjing, Jiangsu, China
`yanhuili@nju.edu.cn`

**Tianyi Zhang**
Xidian University
Xi'an, China
`22171110624@stu.xidian.edu.cn`

**Diandong Liu**
Shaanxi University of Science & Technology
Xi'an, China
`221612075@sust.edu.cn`

## Abstract

Instruction fine-tuning is a key technique for enhancing the performance of large language models (LLMs), but low-quality and redundant data often hinder its effectiveness. Recent studies suggest that filtering a small amount of high-quality data for instruction fine-tuning can achieve faster and more efficient training performance. However, existing data filtering approaches predominantly depend on predefined evaluation models or manually designed metrics, without leveraging information from the target LLM itself. This limitation may result in a mismatch between the filtering criteria and the actual requirements of the LLM being fine-tuned, thereby reducing the effectiveness of the fine-tuning process. To address these issues, we propose a novel perspective: the hidden states of LLMs implicitly reflect the quality of the training data. Based on this insight, we propose a novel data filtering method that extracts the hidden states that reflect the target LLM's perception of the data as representative features, and builds a data classification model upon them, which outputs the Contrastive Perception Quality Score (CPQS) for dataset filtering. Our experiments are conducted in both general and downstream domains. ① In the general domain, our experiments show that training on under 10% of the data from both the Alpaca_GPT4 and DeepSeek-R1 synthesized reasoning datasets enables our method to outperform models trained on the complete datasets. Moreover, it surpasses the performance of current state-of-the-art data-selection techniques. ② In downstream tasks, our approach delivers an average performance gain exceeding 3.6% over leading data-selection algorithms across multiple benchmarks, including GSM8K, HumanEval, and HumanEval-Plus.

## 1 Introduction

Large language models (LLMs) (Brown et al., 2020; Chiang et al., 2023; Yang et al., 2024a; Zeng et al., 2024), such as ChatGPT (OpenAI, 2023; Ouyang et al., 2022a), have led to a groundbreaking shift in the realm of artificial intelligence in recent years. These models excel in understanding and handling a wide array of complex language tasks. A critical factor behind their success is instruction tuning (Ding et al., 2023; Ouyang et al., 2022b; Sun et al., 2023; Yu et al., 2023), which enables models to follow user instructions accurately and exhibit outstanding performance on multiple downstream tasks (Ren et al., 2024; Sun et al., 2025; Wang et al., 2023a; Zhou et al., 2024).

---

[*]Corresponding author.

During the instruction tuning process, a high-quality training dataset is essential for effective fine-tuning. Early research on creating such datasets relied on expert-designed responses (Khashabi et al., 2020; Ye et al., 2021; Wang et al., 2022), but these efforts were limited by labor and cost constraints. More recent studies have used powerful teacher LLMs to generate data (Lee et al., 2024; Li et al., 2024a; Wang et al., 2023b). The primary issue with these methods is that, in large-scale data generation, the quality of the generated data varies significantly, with both high-quality and low-quality data being produced. Zhou et al. (2023) propose the LIMA model with a new perspective to address this issue: using as few as 1,000 carefully chosen, high-quality instruction examples can substantially enhance model performance. This result suggests that developing practical algorithms to extract **a small, high-quality subset** from large training datasets can lead to improved training outcomes.

Building on this idea, data filtering has become a popular area of research for efficient instruction fine-tuning (Cao et al., 2023b; Chen et al., 2023a; Chiang et al., 2023; Liu et al., 2024d). On one side, a main group of researchers has attempted to use *predefined reward models* (Chen et al., 2024; Lu et al., 2024; Bukharin et al., 2024; Li et al., 2024b) to score data and filter it accordingly. Other studies have analyzed data quality from multiple dimensions (Du et al., 2023; Li et al., 2024c;d; Wu et al., 2023; Yu et al., 2024) and selected data according to *defined quality metrics*. In summary, previous research mainly relies on predefined evaluation models or metrics for data filtering, **without considering information from specific LLMs to be fine-tuned**. This gap could lead to a mismatch between the evaluation criteria and the actual needs of the LLMs being fine-tuned, potentially impacting the success of the fine-tuning process.

To address this gap, this study uses runtime information from large language models as features to enhance data representation. By working with feature vectors derived from both high-quality and low-quality data processed by LLMs, it constructs a data classification model that helps select better-suited, higher-quality data for fine-tuning LLMs, making the process more effective and efficient. Specifically, our approach relies on two key ideas.

(a) **Employ hidden states as LLM features**: we extract the hidden states (i.e., neuron activations) (Goloviznina & Kotelnikov, 2024; Wang et al., 2024a) of the target LLM as representative features, which encode the model's implicit evaluation of data quality. Leveraging them enables us to analyze training data quality from the LLM's own perspective.

(b) **Label training data based on quality tiers**: we build datasets with high-quality and low-quality labels (Wettig et al., 2024; Wen et al., 2024), enabling contrastive training that allows our CNN model, trained on LLM hidden states, to more effectively capture and interpret the implicit evaluation differences that the LLM encodes regarding data quality.

To realize the above ideas, our method is divided into the following four steps: ① We first construct an instruction fine-tuning dataset with varying performance, containing both "high-quality" and "low-quality" samples; ② we then extract the hidden states of the target fine-tuning model for each instruction; ③ based on these hidden states, we train a Convolutional Neural Network (CNN) model to identify whether the current testing sample is effective (i.e., of high quality) or not; ④ during the prediction phase, the CNN model analyzes the hidden states perceived by the LLM for each instruction, generating a prediction probability and classification result. The prediction probability classified as effective is referred to as $\mathcal{CPQS}$, which serves as the criterion for dataset filtering.

Through extensive experiments, we validated that our algorithm performs excellently in data selection for general and downstream domain tasks. In the general domain, we tested using the Alpaca_GPT4 datasets (Taori et al., 2023) and the reasoning-deepseek dataset (Hartford & Computations, 2025). The experimental results show that the selected data amount by our method was less than 10% of the original dataset, yet it outperformed models trained on the entire dataset. Additionally, our algorithm is proven to surpass various state-of-the-art algorithms on multiple LLMs. In downstream task domains, such as mathematical problems and programming tasks, the experimental results show that our algorithm outperformed existing state-of-the-art algorithms by an average of 3.6 percentage points on benchmark tests like GSM8K, HumanEval (Chen et al., 2021), and HumanEval-Plus (Liu et al., 2024c) with the same data scale.

The main contributions of this paper can be summarized as follows:

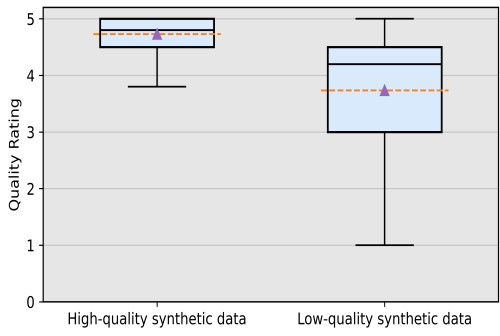

Figure 1: Synthetic-data scored by ALPAGA-SUS

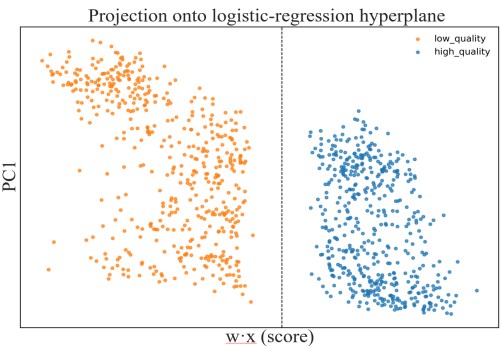

Figure 2: Quality Separation in Hidden-State Space: Logistic-Projection $\mathbf{w} \cdot \mathbf{x}$ vs. PC1

**Method** We proposed an efficient and accurate data selection method based on the LLM's own contrastive perception quality score, significantly enhancing instruction-tuning performance.

**Study** This paper presents extensive empirical studies that utilize two general fine-tuning datasets and two task-specific datasets. The results indicate that the proposed data selection method achieves optimal performance in both general tasks and specific areas such as mathematics and programming.

## 2 MOTIVATION

We introduce our core idea: using LLM hidden states to extract signals of training data quality, illustrated with an initial experiment.

Recent studies show that LLM hidden states hold rich, actionable information. Xie et al. (2022) measured task specialization through variability within and between classes in hidden states; Servedio et al. (2025) evaluated factual accuracy from hidden-state signals; and Qian et al. (2025) used hidden states to actively filter harmful inputs.

Motivated by these findings, we hypothesize that LLM hidden states encode data-quality signals. To test this, we conduct an initial experiment and use models of different parameter scales to generate high- and low-quality data (the rationale behind this selection strategy will be discussed in detail in Section 3.1). Specifically, we use GPT-4–produced *Alpaca_GPT4* (Taori et al., 2023) as high-quality data, while low-quality data are obtained by regenerating samples with smaller models (e.g., Llama-3.2-1B-Instruct, Qwen2.5-1.5B-Instruct). Following Chen et al. (2024), we use DeepSeek-V3 (Liu et al., 2024a) to assign a quality score to each sample. As shown in Fig. 1, the high-quality set averages 4.73 (mostly 4.5–5.0) versus 3.73 for the low-quality set (mostly 3.0–4.5), indicating a clear gap.

We select the top 500 scored samples from the high-quality set and the bottom 500 from the low-quality set. Using the last-layer hidden states of Qwen2.5-7B-Instruct as embeddings, we train a linear logistic regressor; stratified 5-fold CV yields AUC = 1.00 (mean ± std), indicating linear separability. For visualization, we project each embedding $\mathbf{x}$ onto the hyperplane normal $\mathbf{w}$ to obtain $\mathbf{w} \cdot \mathbf{x}$, plot it against PC1, and mark the decision boundary $\mathbf{w} \cdot \mathbf{x} + b = 0$; colors denote ground-truth labels and show clear separation along $\mathbf{w} \cdot \mathbf{x}$ (Fig. 2). Overall, the two sets are linearly separable in the model's hidden-state space.

Building on these findings, we discover that **LLM hidden states differentiate high- from low-quality data**. We train a CNN using hidden-state embeddings from both sets and leverage its outputs to score new samples, providing a prompt-insensitive evaluation of training data quality.

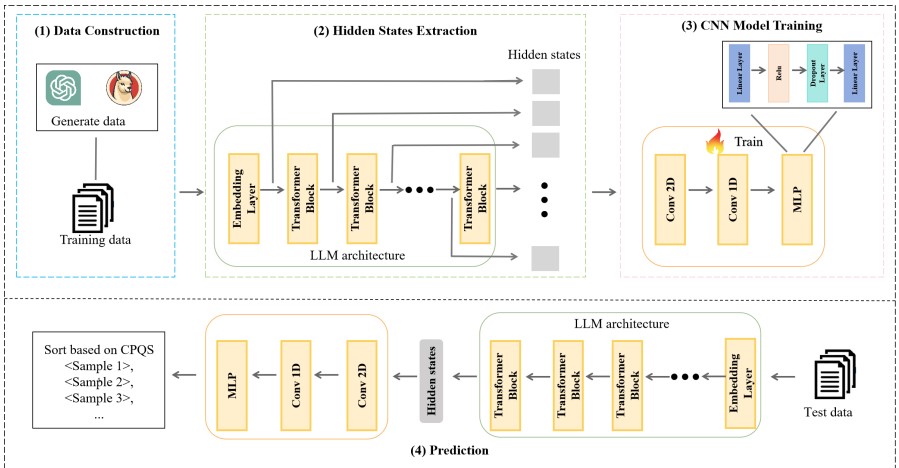

Figure 3: Overall algorithm architecture diagram

# 3 PROPOSED METHODS

This section provides a detailed description of our method. The core of our approach is to train an external CNN model by extracting the hidden states perceived by an LM on high-quality and low-quality general fine-tuning datasets for each data point. This model will be used to analyze the implicit evaluation of the training data within the hidden states of the LLM. The method consists of four key steps, as shown in Figure 3. We will discuss each step in depth and analyze it accordingly.

## 3.1 CONSTRUCTION OF HIGH- AND LOW-QUALITY DATASETS

Prior work has demonstrated that data synthesized by stronger models, such as GPT-4, tends to be of higher quality and can substantially improve the downstream performance of smaller models (Wang et al., 2023b; Peng et al., 2023). Motivated by this observation, we treat outputs from powerful models (e.g., GPT-4) as high-quality data, while the low-quality set is synthesized using much smaller models. To construct the training data for the CNN, we randomly sampled 5,000 items from the Alpaca_GPT4 dataset as high-quality samples, capping the subset to mitigate overfitting risks. The Alpaca_GPT4 dataset itself is a large-scale, general-domain fine-tuning corpus containing 52,000 entries, each represented as a triplet ⟨Instruction, Input, Response⟩, where Instruction specifies the task, Input provides auxiliary context, and Response is the GPT-4–generated answer.

To construct the *low-quality data samples*, we used two small LMs—Llama-3.2-1B-Instruct (Dubey et al., 2024) and Qwen2.5-1.5B-Instruct (Yang et al., 2024b). Specifically, (1) from each Alpaca_GPT4 entry, we extracted the pair ⟨Instruction, Input⟩; (2) we then used the two models to generate the corresponding Response, forming new triplets ⟨Instruction, Input, Response⟩; and (3) we uniformly sampled 5,000 entries from each model's outputs, resulting in 10,000 items as the low-quality dataset.

Finally, we combine the high-quality and low-quality datasets to obtain the complete training set.

## 3.2 EXTRACTION OF HIDDEN STATES

For the collected training dataset, we concatenate the Instruction and Input parts of each entry and use this concatenated value as the "user" input to the model, while the Response part is used as the "assistant" input. This combined entry is then passed to the model to obtain the hidden states across all layers of the model. We retain only the hidden state corresponding to the Response part of each entry. This choice is made because the evaluation of the fine-tuning dataset's quality primarily depends on the quality of the Response.

### 3.3 TRAINING OF THE EXTERNAL CNN MODEL

The core idea of our algorithm is that the hidden states generated by the LLM contain an implicit evaluation of the quality of the training data. To analyze the LLM's quality assessment of the training data, we propose using an external model to interpret the hidden state vectors produced by the LLM. The external model uses a Convolutional Neural Network (CNN) architecture. We employ a simple CNN to learn information relevant to data-quality assessment from an LLM's hidden states (subsequent experiments show that this CNN performs quite well compared with simple models such as MLP), employing both 2D and 1D convolutions to capture detailed spatial and sequential patterns within the hidden-state vectors.

The CNN model further employs adaptive max pooling and fully connected layers to perform binary classification, distinguishing between high- and low-quality data based on the LLM's hidden states. For efficient training, we optimize the model using the Adam optimizer with a learning rate of 0.0001 and employ gradient accumulation alongside mixed precision training (AMP) to reduce memory overhead and accelerate convergence. The training process minimizes CrossEntropyLoss, with periodic checkpoints saved to ensure robustness. To guarantee optimal performance, we track the loss trajectory during training and retain the best-performing model based on validation metrics.

During the training process, we use the hidden states of each sample obtained in the previous section, along with their corresponding positive and negative labels, for training. The positive-to-negative sample ratio was set to 1:2, yielding a total of 15,000 samples. We chose this ratio because it accelerates loss reduction, and with 15,000 training examples, the loss had already nearly converged. Inspired by contrastive learning, we framed the CNN's training as a binary classification task. This enables the model to distinguish between the different information perceived by the hidden states of the LLM for good and bad samples. In doing so, the CNN model can more accurately reflect the LLM's evaluation of the training data quality.

### 3.4 PREDICTION OF CONTRASTIVE PERCEPTION QUALITY SCORE

In the prediction phase, we introduce the Contrastive Perception Quality Score ($\mathcal{CPQS}$) to evaluate the training quality of each instruction-following sample. A higher $\mathcal{CPQS}$ indicates that the LLM assigns greater importance to the data, implying better training effectiveness and higher quality. The calculation process is as follows:

1. For each instruction-following sample, we first concatenate the Instruction and Input as the "user" input and the Response as the "assistant" input. The entry is then fed into the LLM, from which we extract only the hidden states corresponding to the Response part. These hidden states are passed into a pre-trained CNN to predict class probabilities. We focus on the probability of class 1, which indicates how likely the LLM regards the sample as high-quality. This value, denoted as $\mathcal{CPQS}$, serves as our data quality metric. The calculation is as follows:

$$\mathcal{CPQS}_i = p_i = f(\mathbf{x}_i),$$

where $p_i$ is the predicted probability for the $i$-th sample, representing the likelihood that the sample belongs to the positive class, and $f(\mathbf{x}_i)$ is the output of the LLM's hidden state vector for the $i$-th entry, processed by the CNN model to produce the probability that the sample belongs to class 1.

2. After calculating the $\mathcal{CPQS}$ for all samples, we sort the entire dataset based on these probabilities in descending order and select the top $K$ samples for further processing. The specific selection process is represented as:

$$\mathcal{D}_{\text{selected}} = \text{top}_K \left( \{\mathcal{CPQS}_i\}_{i=1}^N \right),$$

where $\mathcal{D}_{\text{selected}}$ is the set of the top $K$ selected samples from the dataset based on their $\mathcal{CPQS}$ values, and $\mathbf{top}_K$ denotes selecting the top $K$ samples after sorting.

## 4 EXPERIMENTAL SETUP

### 4.1 DATASETS AND MODELS

We benchmark our method on four datasets—two general-domain and two downstream—and three 7 B-parameter open-source LLMs.

**Training Datasets.** ① *General domain.* (i) **Alpaca_GPT4** (Taori et al., 2023): 52,K instruction–response pairs generated by GPT-4, widely used as a standard benchmark dataset; overall medium quality with relatively fluent but sometimes shallow responses. (ii) **Reasoning-DeepSeek**: 146,K long-chain-reasoning samples distilled from the 300,K Dolphin–R1 corpus (Hartford & Computations, 2025) after filtering out sequences longer than $2,048$ tokens; considered high quality due to their complexity and coherence, particularly suitable for evaluating reasoning ability. ② *Downstream tasks.* (i) **GSM8K** Cobbe et al. (2021): 7.5,K training and 1,K test elementary-math word problems, designed to assess arithmetic and step-by-step reasoning. (ii) **Magicoder-Evol-Instruct-110K** (Wei et al., 2024): 110,K programming instructions covering a wide range of languages and problem types, offering a challenging benchmark for code generation and instruction following.

**Models.** (i) **Llama 2–7B** (Touvron et al., 2023): a base model with 7B parameters, pretrained for general natural language understanding and generation. (ii) **Llama 2–7B–Chat** (Touvron et al., 2023): a dialogue-tuned variant of Llama 2–7B, optimized for multi-turn conversational scenarios. (iii) **Qwen2.5–7B–Instruct** (Yang et al., 2024b): an instruction-tuned model designed for text and code generation, mathematical reasoning, and complex multi-step tasks, providing strong performance across diverse downstream benchmarks.

## 4.2 COMPARISON ALGORITHMS

To validate our algorithm, we compared it with three state-of-the-art data selection methods that have received broad attention (ICLR 2024, ACL 2024, arXiv preprint):

1. **ALPAGASUS**: Chen et al. (2024) leveraged LLMs such as ChatGPT to automatically detect and filter out low-quality data. Specifically, it employs a prompt-based scoring system where a strong model evaluates instruction-response pairs and retains only those exceeding a predefined quality threshold.

2. **MoDS**: Du et al. (2023) proposed a data selection strategy based on the criteria of quality, coverage, and necessity. This approach sequentially applies a reward model to extract high-quality instances, clustering algorithms to ensure diverse instruction coverage, and target-model evaluation to select instances that address specific capability gaps.

3. **Superfiltering**: Li et al. (2024b) introduced a method that uses a smaller model to filter data by instruction-following difficulty before fine-tuning a larger model. By demonstrating that weak models share similar perceptions of data difficulty with strong models, this weak-to-strong paradigm significantly reduces the computational overhead of the filtering process.

## 4.3 IMPLEMENTATION DETAILS

We conducted our experiments on a platform equipped with two NVIDIA RTX 4090 GPUs. We adopted LoRA-based fine-tuning using the Llama-Factory framework (Zheng et al., 2024). During supervised fine-tuning (SFT), we used `bf16` precision, three epochs, a learning rate of 5e-5, a batch size of 16, and a maximum sequence length of 2048. The LoRA scaling factor was $\alpha = 8$, and the rank was $r = 16$. For the deployment and inference of our model, we utilized vLLM (Kwon et al., 2023). During inference, we configured the temperature to 0, maintained the precision at bf16, and set the maximum sequence length to 2048.

## 4.4 EVALUATION METRICS

**General Domain Evaluation Standards.** We design evaluation metrics tailored to each data type. For Alpaca_GPT4, we adopt three common metrics: ① *Pair-wise Comparison*, following Chen et al. (2024), where GPT-4o scores model outputs on Koala (180), WizardLM (218), Self-instruct (252), and Vicuna (80) across relevance and accuracy (1–10 scale), with two rounds to mitigate position bias and results categorized as Win/Tie/Loss; ② *Open LLM Leaderboard*, benchmarking on MMLU, ARC, HellaSwag, and TruthfulQA via the lm-evaluation-harness (Gao et al., 2024) (batch size 8); and ③ *Alpaca Eval*, measuring GPT-4o win rate against text-davinci-003 (Li et al., 2023). For the reasoning-deepseek dataset, we evaluate on GSM8K, Math_500, HumanEval, and GPQA, covering mathematical reasoning and code generation, using lm-evaluation-harness (Gao et al., 2024) for GSM8K and HumanEval, and EvalScope (Team, 2023) for Math_500 and GPQA.

Table 1: Comparative evaluation of data-selection algorithms on Alpaca_GPT4 with varying sample sizes across MMLU, ARC, TruthfulQA, HellaSwag, and AlpacaEval.

| Size | Algorithm | MMLU | ARC | TruthfulQA | HellaSwag | Average | AlpacaEval |
|------|-----------|------|-----|------------|-----------|---------|------------|
| 1k | Self | 42.07 | 45.73 | 45.54 | 77.38 | **52.68** | 55.98 |
| 1k | Superfiltering | 41.54 | 46.67 | 44.59 | 77.33 | 52.53 | 55.87 |
| 1k | MoDs | 40.21 | 46.25 | 46.79 | 77.20 | 52.61 | 52.78 |
| 1k | Alpagasus | 40.32 | 46.76 | 43.50 | 77.12 | 51.92 | 49.65 |
| 2k | Self | 44.30 | 47.18 | 45.68 | 77.54 | **53.68** | 57.90 |
| 2k | Superfiltering | 42.42 | 47.61 | 45.77 | 77.50 | 53.32 | 57.02 |
| 2k | MoDs | 43.21 | 47.35 | 45.78 | 77.59 | 53.48 | 53.42 |
| 2k | Alpagasus | 42.93 | 46.42 | 45.65 | 77.71 | 53.18 | 56.90 |
| 3k | Self | 43.87 | 47.89 | 46.03 | 77.69 | 53.87 | 58.76 |
| 5k | Self | 44.33 | 48.21 | 47.42 | 77.83 | **54.40** | **59.94** |
| 12k | Alpagasus | 43.22 | 48.29 | 46.86 | 78.43 | 54.20 | 58.80 |
| 52k | Full | 42.15 | 48.23 | 48.46 | 78.65 | 54.37 | 59.81 |

**Evaluation Criteria for Downstream Task Domains.** For downstream evaluation, we target two domains—mathematical reasoning and code generation. In mathematics, we use the GSM8K dataset (Cobbe et al., 2021)—a set of 1,000 middle- and high-school arithmetic, algebra, and geometry problems—to measure problem-solving and reasoning skills. For code generation, we employ the 164-question HumanEval benchmark (Chen et al., 2021) and its more demanding extension, HumanEval-Plus (Liu et al., 2023; 2024c), which adds complex tasks to assess code accuracy, reasoning, adaptability, and robustness across diverse inputs.

## 5 EXPERIMENTAL RESULT

### 5.1 GENERAL DOMAIN EVALUATION

In this section, we compare our data-selection algorithm with three state-of-the-art methods. Experiments are conducted on Alpaca_GPT4 with Llama2-7B and Reasoning-DeepSeek with Qwen2.5-7B-Instruct to evaluate performance across models and datasets. ① On Alpaca_GPT4, we train with subsets of 1K, 2K, and larger sizes, evaluating on MMLU, ARC-Challenge, TruthfulQA, HellaSwag, and AlpacaEval (Table 1). ② On Reasoning-DeepSeek, we use 10k, 20k, and 50k samples, benchmarking against the base model, Superfiltering, and ALPAGASUS; MoDS fails on larger subsets. Results are summarized in Tables 1 and 2.

**Our algorithm outperforms state-of-the-art methods and achieves better results than full-dataset training on the Alpaca_GPT4 dataset with Llama2-7B, using less than 10% of the data.** Table 1 presents a performance comparison between our algorithm and other methods on models trained with filtered subsets of varying sizes from the Alpaca_GPT4 dataset. As shown in the first part of Table 1 (data size = 1k), our approach consistently outperforms all competitors in the low-data regime. With only 1k training instances, it attains a macro-average accuracy of 52.68% over the four public benchmarks and an AlpacaEval win-rate of 55.98%, exceeding the strongest baseline by 0.15 and 0.11 percentage points, respectively. As illustrated in the second part of Table 1 (data size = 2k), doubling the budget to 2k further lifts the macro-average to 53.68% and the AlpacaEval score to 57.90%, while still maintaining a clear margin over all baselines. We additionally conduct pairwise preference tests with GPT-4o on Vicuna-style prompts; the comparison in Figure 4 shows that our 1k model already surpasses the model trained on the full 52k corpus. To assess scalability, we increase the subset size to 3k and 5k. As shown in the third part of Table 1, performance improves steadily: at 5k, the open-domain average reaches 54.40%, and the AlpacaEval win-rate climbs to 59.94%. Notably, the 5k model outperforms the Alpagasus-filtered 12k model and surpasses the full 52k model on every metric, confirming the superior efficiency of our selection criterion.

**Our algorithm also outperforms existing methods on the Reasoning-DeepSeek dataset with the Qwen2.5-7B-Instruct model, achieving better performance using less than 10% of the data compared to full-dataset training.** As shown in Table 2, despite the generally higher quality of the Reasoning-DeepSeek dataset, existing algorithms fail to achieve outstanding performance, while

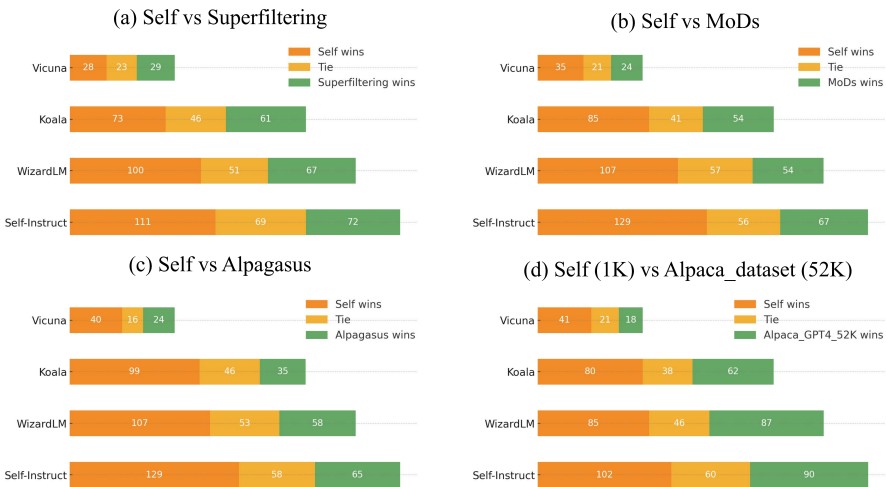

Figure 4: Performance Comparison of Data Selection Methods on the Llama2-7B Model Using the Alpaca_GPT4_Dataset.

Table 2: Performance of models trained with data filtering on the Reasoning-DeepSeek dataset across GSM8K, Math_500, HumanEval, and GPQA benchmarks.

| Size | Model | GSM8K | Math_500 | HumanEval | GPQA | Average |
|---|---|---|---|---|---|---|
| — | Base | 76.27 | 73.40 | 64.02 | 30.30 | 61.00 |
| 10k | Self | 85.37 | 72.40 | 67.68 | 36.36 | **65.45** |
| 10k | Superfiltering | 81.43 | 70.40 | 62.20 | 31.82 | 61.46 |
| 20k | Self | 85.14 | 76.20 | 64.02 | 34.85 | **65.05** |
| 20k | Superfiltering | 76.19 | 71.60 | 63.41 | 28.79 | 60.00 |
| 50k | Self | 84.99 | 74.20 | 64.02 | 36.87 | **65.02** |
| 113k | Alpagasus | 84.84 | 66.52 | 64.63 | 28.18 | 61.04 |
| 146k | Full | 85.06 | 70.60 | 57.32 | 30.71 | 60.92 |

our method consistently leads. Specifically, the model trained on our 10K filtered subset achieved an average score of 65.45 across mathematical and coding reasoning tasks, surpassing models trained on Superfiltering's 10K and 20K datasets, Alpagasus's 113K dataset, and even the full 146K dataset. Notably, the performance on Math_500 initially declined compared to the base model. We attribute this to a limited maximum generation length setting of 8K tokens; our additional experiments confirmed that increasing the token limit enhances performance significantly. Moreover, we observed that increasing the size of the filtered training dataset beyond a certain point did not yield substantial improvements. For example, the model trained on the 10K dataset outperformed both the 50K filtered dataset and the full 146K dataset, with the 10K model surpassing the full 146K-trained model by 3.48 points. This indicates potential noise and redundancy in the full dataset, underscoring the effectiveness of our targeted filtering approach.

In addition to the evaluations mentioned, we applied our algorithm to select 10k samples from the 930k Tulu3-SFT-Mixture dataset (Appendix A.2), where it continued to outperform competing methods. We also explored its performance under full-parameter fine-tuning, with results in Appendix A.1, confirming its superiority. The iterative nature and performance-influencing factors of our method are analyzed in Appendices A.3 and A.4. Further, we demonstrate the generalization of our method across different LLM sizes in Appendix A.5, compare its performance with models using different architectures for hidden-state extraction in Appendix A.6, and present a comparison of performance and efficiency with other methods in Appendix A.7.

## 5.2 DOWNSTREAM TASK EVALUATION

We evaluated our algorithm on downstream-task datasets using Llama2-7B-Chat and Qwen2.5-7B-Instruct. Following the practice of fine-tuning pre-optimized models, we tested on GSM8K (Cobbe

Table 3: Performance Evaluation of Models Trained with Different Data Selection Methods on the GSM8K Dataset in the Mathematics Domain

| Model | Original | Self | MoDs | Alpagasus | Superfiltering | Full |
|---|---|---|---|---|---|---|
| Llama 2-7B-Chat | 24.56 | 25.25 | 23.05 | 23.12 | 17.06 | **35.56** |
| Qwen 2.5-7B-Instruct | 76.27 | **81.12** | 79.45 | 76.27 | 76.80 | 69.83 |

Table 4: Performance Evaluation of Models Trained with Different Algorithms on Llama 2-7B-Chat in the Code Domain (pass@1).

| | Original | Self | MoDs | Alpagasus | Superfiltering |
|---|---|---|---|---|---|
| HumanEval | 13.4 | **16.5** | 12.2 | 12.2 | 10.0 |
| HumanEval-Plus | 11.6 | **13.4** | 9.8 | 10.4 | 9.1 |
| Average | 12.5 | **14.95** | 11 | 11.3 | 9.55 |

et al., 2021) (math) and Magicoder-Evol-Instruct-110K (Wei et al., 2024) (code), filtering them to 500 and 1,000 samples respectively. Performance was assessed on GSM8K test, HumanEval, and HumanEval-Plus. For ALPAGASUS, GPT-4o-mini was used for data selection due to its higher efficiency and lower cost.

**Our algorithm outperforms other algorithms in the field of mathematics.** Table 3 presents results on the GSM8K Dataset, where our algorithm outperforms the others by 4.17 points on Llama 2-7B-Chat and 3.61 points on Qwen 2.5-7B-Instruct. Notably, the model trained on the full GSM8K dataset underperforms on Qwen 2.5-7B-Instruct (score 76.27), while the model trained on 500 selected data points achieves 81.12, surpassing the full dataset by 11.29 points. This demonstrates the effectiveness of our algorithm in the field of mathematics.

**Our algorithm outperforms other algorithms in the field of code.** Tables 4 and 5 show results with the Magicoder-Evol-Instruct-110K dataset. On Llama 2-7B-Chat (Table 4), our algorithm outperforms others by 4.3 points on average, and models trained on 1000 data points from other algorithms performed worse than the original. In contrast, the model trained on 1000 points selected by our algorithm improved by 2.45 points. On Qwen 2.5-7B-Instruct (Table 5), the model trained with our selected data outperformed others by 1.47 points on average. However, all models trained on the filtered Magicoder-Evol-Instruct-110K dataset showed performance degradation, likely due to its lower quality for this model. This demonstrates the effectiveness of our algorithm in the code domain.

## 5.3 THE IMPACT OF DIFFERENT HIDDEN LAYERS ON MODEL PERFORMANCE

To investigate the impact of different hidden layers on data selection performance, we trained separate CNN models using hidden states from various layers of Qwen 2.5-7B-Instruct. Specifically, we extracted hidden states from the first 9 layers, the middle 9 layers, the last 11 layers, and the final layer. We then evaluated the filtering performance of these layer-specific models on the GSM8K dataset (Cobbe et al., 2021) (mathematics) and the Magicoder-Evol-Instruct-110K dataset (Wei et al., 2024) (code).

**Utilizing hidden states from all layers yields the optimal data filtering performance.** Table 6 presents the comparison of CNN models trained on different hidden layer segments. Across both mathematical and coding tasks, the model trained with hidden states from all layers (Full) consistently achieves the highest performance. When utilizing subset layers, the optimal segment varies by task. On the GSM8K dataset, the early layers (first 9) perform best among the subsets with a score of 84.23. Conversely, in the code domain (evaluated via HumanEval and HumanEval-Plus), the later layers (last 11) demonstrate superior subset performance, achieving an average score of 76.55. These results indicate two main points: first, while later hidden layers capture critical task-specific information (such as for code generation), different tasks may rely on distinct layer representations; second, utilizing any individual subset of layers still lags behind the comprehensive representation provided by all layers combined.

Table 5: Performance Evaluation of Models Trained with Different Algorithms on Qwen 2.5-7B-Instruct in the Code Domain.

|  | Original | Self | MoDs | Alpagasus | Superfiltering |
|---|---|---|---|---|---|
| HumanEval | **82.9** | 80.0 | 78.7 | 78.0 | 79.3 |
| HumanEval-Plus | **75.6** | 74.4 | 72.6 | 72.6 | 73.2 |
| Average | **79.25** | 77.2 | 75.65 | 75.3 | 76.25 |

Table 6: Comparison of CNN Model Performance Trained on Different Hidden Layer Parts of Qwen 2.5-7B-Instruct.

| Dataset / Metric | Full | Early (9) | Middle (9) | Last (11) | Final (1) |
|---|---|---|---|---|---|
| GSM8K | **84.91** | 84.23 | 83.85 | 83.70 | 83.70 |
| HumanEval (pass@1) | **80.0** | 75.0 | 77.4 | 79.3 | 78.7 |
| HumanEval-Plus (pass@1) | **74.4** | 70.1 | 71.3 | 73.8 | 72.6 |

## 6 RELATED WORK

**Data Selection Strategies.** The goal of instruction tuning (Liu et al., 2024b; Longpre et al., 2023; Sanh et al., 2022; Wei et al., 2022) is to help LLMs better understand human task requirements. Early research primarily focused on building large-scale instruction datasets, but studies like LIMA have shown that only a small amount of high-quality data is needed during instruction fine-tuning to achieve good results. Existing data selection methods can be classified into four categories: indicator-based methods, trainable LLM-based methods, powerful LLM-based methods, and small-model-based methods.

Indicator-based methods use a metric system to identify multiple evaluation indicators to comprehensively assess data quality (Cao et al., 2023a; Wei et al., 2023). Trainable LLM-based methods treat the LLM as a trainable data selector, processing and assigning scores to each instruction fine-tuning data for selection (Chen et al., 2023b; Li et al., 2024c). Powerful LLM-based approaches, such as those using models like ChatGPT, typically design prompt templates and leverage the model's capabilities to quantitatively evaluate the quality of data samples (Chen et al., 2024; Liu et al., 2024d). Finally, small-model-based methods involve using external small models to score the data or projecting the data samples into vectors with a small model for further processing and selection (Chen et al., 2023a; Li et al., 2024b).

**Performance Evaluation of LLMs.** The evaluation of LLMs is typically done through automatic evaluation, human evaluation, and using LLMs as evaluators. Automatic evaluation relies on pre-defined criteria and quantitative assessment (Chen et al., 2021; Hendrycks et al., 2021; Wang et al., 2024b). Human evaluation focuses on qualitative aspects like clarity, consistency, and factual accuracy, and is essential for quality assessment (van der Lee et al., 2021; Zheng et al., 2023). However, due to its time and labor demands, using powerful LLMs (Chen et al., 2024; Huang et al., 2024) to evaluate other LLMs has become a popular approach in recent years.

## 7 CONCLUSION

This paper addresses the issue of low-quality and redundant data in LLM instruction fine-tuning. Based on the hidden states that reflect the target LLM's perception of the data, we build a data classification model and define CPQS as its output. Using CPQS as the criterion, our method filters high-quality data subsets, thereby improving the efficiency and effectiveness of instruction fine-tuning.

Experimental results show that our approach achieves superior performance with less than 10% of the original data compared to models trained on the full dataset. It also outperforms existing data selection techniques at equal data scales. In downstream tasks, such as mathematical problem solving (GSM8K) and programming (HumanEval, HumanEval+), our method provides a 3.6% average performance improvement over current state-of-the-art data selection algorithms. The code and data have been provided online at `https://github.com/renllll/CPQS-Tuning`.

ACKNOWLEDGMENTS

The authors thank anonymous reviewers for their time and invaluable feedback to improve our study. This work is supported by the National Natural Science Foundation of China (Grant Nos. 62172202, 62272221, and 62572226).

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

## A APPENDIX

### A.1 FULL-SCALE FINE-TUNING EXPERIMENT

At the 1k and 2k data scales, we performed full-parameter fine-tuning on the datasets produced by each selection method to assess their performance under more exhaustive training. The setup employed two H20 GPUs, a learning rate of 1e-5, a batch size of 64, and three training epochs. Results are summarized in Table 7.

After full fine-tuning, every method achieved noticeably higher scores on all five benchmarks than with LoRA-only tuning, demonstrating that updating the entire set of model weights can further

Table 7: Comparative evaluation of models trained with different data-selection algorithms and sample sizes (1k vs. 2k) on five benchmarks.

| Size | Model | MMLU | ARC | Truthful-QA | HellaSwag | Avg. | AlpacaEval |
|------|-------|------|-----|-------------|-----------|------|------------|
| 1k | Self | 41.95 | 47.87 | 50.92 | 77.57 | **54.58** | **65.93** |
| 1k | Superfiltering | 41.76 | 47.78 | 51.05 | 77.56 | 54.54 | 64.36 |
| 1k | MoDs | 41.43 | 47.53 | 48.46 | 76.82 | 53.56 | 57.02 |
| 1k | Alpagasus | 39.76 | 47.87 | 52.08 | 77.54 | 54.31 | 55.14 |
| 2k | Self | 42.06 | 48.25 | 53.47 | 77.66 | **55.36** | **69.73** |
| 2k | Superfiltering | 42.14 | 48.63 | 51.71 | 77.97 | 55.11 | 69.69 |
| 2k | MoDs | 39.94 | 46.67 | 53.10 | 77.67 | 54.34 | 54.10 |
| 2k | Alpagasus | 41.66 | 47.78 | 51.42 | 76.66 | 54.38 | 64.16 |

unlock data value. Notably, the dataset selected by our algorithm led in both average score and AlpacaEval at both the 1k and 2k scales, confirming its clear performance advantage.

## A.2 ADDITIONAL EXPERIMENTS ON DATASET FILTERING EFFECTIVENESS

We conducted additional experiments on the Llama2-7B model using the TULU-3-SFT-mixture dataset, comparing results from filtering 930K samples down to 10K samples. As shown in Table 8, our method outperforms other algorithms on both open-ended large model evaluation benchmarks and AlpacaEval (notably, the MoDs algorithm was excluded from comparison due to GPU memory limitations with larger sample sizes). Our approach achieved an average score of 53.27 on the open-ended evaluation benchmarks and 52.37 on AlpacaEval.

Table 8: Comparative Evaluation of Llama2-7B Models Trained with Different Data Selection Algorithms on TULU-3-SFT-mixture (Filtered to 10K Samples) Across MMLU, ARC (Challenge), TruthfulQA, HellaSwag, and AlpacaEval Benchmarks.

| Model | MMLU | ARC | TruthfulQA | HellaSwag | Avg. | AlpacaEval |
|-------|------|-----|------------|-----------|------|------------|
| Self | 44.54 | 45.05 | 46.42 | 77.08 | **53.27** | 52.37 |
| Superfiltering | 42.98 | 45.56 | 43.49 | 76.87 | 52.23 | 45.90 |
| Alpagasus | 42.95 | 46.44 | 44.42 | 76.39 | 52.55 | 32.19 |

We further evaluated pairwise model comparisons using AlpacaEval to measure the win rates between different approaches. As demonstrated in Table 9, our method shows significant advantages over other dataset filtering techniques.

Table 9: Pairwise Model Comparison Results (AlpacaEval)

| Model Comparison | Win Rate (%) | Loss Rate (%) |
|------------------|--------------|---------------|
| Self vs Superfliting | **52.25** | 47.74 |
| Self vs Alpagasus | **68.36** | 31.49 |

## A.3 EXPERIMENT ON ITERATIVE MODEL TRAINING AND DATA SELECTION

We conducted experiments to assess the iterativeness of our algorithm. First, we trained a CNN model using our algorithm on the Qwen2.5-7B-Instruct dataset. This trained model was then used to predict and rank the Alpaca_GPT4 dataset. Based on the ranking, we extracted the top 5,000 and the last 10,000 data samples. We then retrained another CNN model using this subset to further filter 1,000 samples from the Alpaca_GPT4 dataset for additional training. As shown in Figure 5, the performance of the newly trained model demonstrated a significant improvement over the previous version. For example, the green sections in the figure represent the number of wins by the newly trained model, which show a clear advantage across multiple datasets.

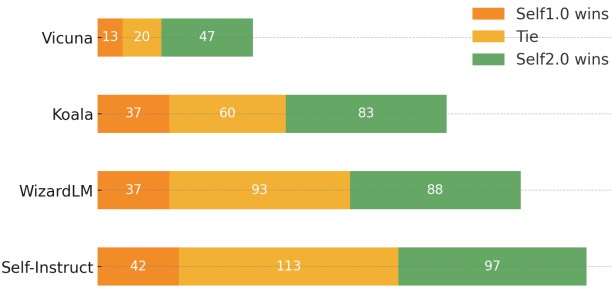

Figure 5: Comparison of Model Performance After Two-Stage Data Selection.

### A.4 PREFERENCES OF DIFFERENT MODELS FOR HIGH-QUALITY DATASETS

In this section, we explore whether different LLMs have distinct preferences for high-quality datasets. To this end, we trained the models on high-quality datasets selected from the GSM8K dataset and Magicoder_Evol_Instruct-110K Dataset using Llama 2-7B-Chat and Qwen 2.5-7B-Instruct. We then compared the performance of these models when exchanging datasets. Specifically, we trained Qwen 2.5-7B-Instruct and Llama 2-7B-Chat on each other's selected datasets and compared their performance with training on their own selected datasets.

As shown in Tables 10 and 11, whether in the mathematical or coding domains, the models trained after swapping datasets did not perform as well as those trained on their original datasets. For both Llama 2-7B-Chat and Qwen 2.5-7B-Instruct, the high-quality dataset considered by one model did not yield the same results when used by the other model. Therefore, our experiment shows that different models exhibit significant differences in selecting high-quality datasets, with each model having its own definition of what constitutes a "high-quality dataset."

Table 10: Performance of Llama 2-7B-Chat and Qwen2.5-7B-Instruct Models Trained on Their Own and Swapped Datasets on the GSM8K Dataset (Cobbe et al., 2021).

| Training Method | Llama 2-7B-Chat | Qwen2.5-7B-Instruct |
|---|---|---|
| Self Training | **25.25** | **84.91** |
| Dataset Swapping | 23.58 | 83.24 |

Table 11: Performance of Llama 2-7B-Chat and Qwen2.5-7B-Instruct Models Trained on Their Own and Swapped Datasets on the HumanEval Dataset.

| Training Method | HumanEval pass@1 | HumanEval-Plus pass@1 | Average |
|---|---|---|---|
| Llama 2-7B-Chat (Self) | **16.5** | **13.4** | **14.95** |
| Llama 2-7B-Chat (Swapped) | 11.2 | 11.0 | 11.1 |
| Qwen2.5-7B-Instruct (Self) | **80.0** | **74.4** | **77.2** |
| Qwen2.5-7B-Instruct (Swapped) | 72.0 | 67.1 | 69.55 |

Furthermore, we explored whether large models of different sizes have distinct preferences for high-quality datasets. To this end, we selected a larger model (such as Qwen 2.5-32B-Instruct) and a smaller model (such as BitCPM4-1B), and conducted experiments on the Reasoning-DeepSeek dataset. Specifically, we trained a CNN for each model to predict and filter the dataset, and then swapped the datasets selected by different models, retraining the models and evaluating their performance differences.

The experimental results, as shown in Table 12, indicate that the models performed better when using their own selected datasets. This was true across the GSM8K, Math_500, HuanEval, and GPQA datasets, where the models consistently achieved better average scores. This further validates the differences in how each model defines and selects "high-quality data."

Table 12: Performance of CNN Models Trained on Their Own and Swapped Datasets on Four Datasets: GSM8K, Math_500, HuanEval, and GPQA.

| Training Method | GSM8K | Math_500 | HuanEval | GPQA | Average |
|---|---|---|---|---|---|
| BitCPM4-1B (self) | 38.06 | 33.00 | 54.88 | 28.79 | **38.68** |
| BitCPM4-1B (swapped) | 42.00 | 32.80 | 48.78 | 24.24 | 36.96 |
| Qwen 2.5-32B (self) | 85.52 | 81.40 | 56.10 | 46.46 | **67.36** |
| Qwen 2.5-32B (swapped) | 85.60 | 80.40 | 51.22 | 42.93 | 65.05 |

Table 13: Comparative evaluation of data-selection methods on **BitCPM4-1B** using Reasoning_DeepSeek subsets across GSM8K, Math_500, HuanEval, and GPQA.

| Size | Method | GSM8K | Math_500 | HuanEval | GPQA | Average |
|---|---|---|---|---|---|---|
| 10k | Self | 38.06 | 33.00 | 54.88 | 28.79 | **38.68** |
| 10k | Superfliting | 39.35 | 30.20 | 52.44 | 24.24 | 36.56 |
| 20k | Self | 37.00 | 32.00 | 53.66 | 29.80 | **38.12** |
| 20k | Superfliting | 36.92 | 30.09 | 54.27 | 27.27 | 37.14 |
| 113k | Alpagasus | 37.38 | 30.80 | 53.66 | 25.76 | 36.90 |

## A.5 GENERALIZATION OF CPQS ACROSS ARCHITECTURES AND SCALES

To assess whether **CPQS** generalizes beyond the three LLMs reported in the main paper, we further evaluated it on two additional model families at different parameter scales: a *small* BitCPM4-1B model and a *large* Qwen2.5-32B-Instruct model. In both cases, CPQS was used to filter the Reasoning_DeepSeek corpus into high-quality subsets of varying sizes (10k, 20k, etc.). We compared CPQS (*self*) against two strong data selection baselines, *Superfliting* and *Alpagasus*, and trained models under identical settings per architecture. We report performance on GSM8K, Math_500, HuanEval, and GPQA, along with the simple average.

Across both the 1B and 32B regimes, CPQS-selected data consistently outperformed all baselines at matched subset sizes. On BitCPM4-1B, CPQS delivered the best averages for 10k and 20k subsets, exceeding Superfliting and the larger 113k Alpagasus subset despite using fewer samples. On Qwen2.5-32B-Instruct, CPQS likewise led at 10k and 20k, with stronger averages than Superfliting and the much larger Alpagasus set. These results indicate that CPQS's selection criteria transfer across architectures and scales, and that *quality* can outweigh *quantity* when curating reasoning-focused training data. The detailed experimental results are presented in Table 13 and Table 14.

## A.6 ABLATION ON THE SELECTOR ARCHITECTURE: CNN VS. MLP VS. TRANSFORMER

To further validate the effectiveness of our selector design, we conducted an ablation study comparing the **CNN architecture** used in our method with two alternative designs: a **multi-layer perceptron (MLP)** and a **Transformer**-based selector.

We evaluated all three architectures on the **Reasoning-DeepSeek** dataset (10k samples selected by CPQS). As shown in Table 15, the **CNN-based selector** consistently outperforms both MLP and Transformer variants across GSM8K, Math_500, HuanEval, and GPQA benchmarks, achieving the best overall average. We attribute this performance advantage to CNN's *strong locality bias* and *computational efficiency*, which enable it to extract hierarchical spatial correlations from hidden states and thus improve data-quality discrimination.

## A.7 ADDITIONAL COMPARATIVE EVALUATIONS AND EFFICIENCY ANALYSIS

### A.7.1 COMPARISON WITH RECENT DATA-SELECTION METHODS

We extend our comparative study by including several recent and representative data-selection approaches: **SelectIT**, **DS2**, and **Deita**, which are widely regarded as strong baselines for instruction-tuning data curation. All methods were evaluated on the *Reasoning-DeepSeek* dataset using **10k** selected samples and fine-tuning a **Qwen2.5-7B-Instruct** model under identical conditions (maxi-

Table 14: Comparative evaluation of data-selection methods on **Qwen2.5-32B-Instruct** using Reasoning_DeepSeek subsets across GSM8K, Math_500, HuanEval, and GPQA.

| Size | Method | GSM8K | Math_500 | HuanEval | GPQA | Average |
|------|--------|-------|----------|----------|------|---------|
| 10k | Self | **85.52** | 81.40 | 56.10 | **46.46** | **67.36** |
| 10k | Superfliting | 85.22 | **81.80** | **56.71** | 43.43 | 66.79 |
| 20k | Self | 84.76 | **81.60** | 53.66 | **46.46** | **66.62** |
| 20k | Superfliting | **84.99** | 81.40 | 49.39 | 42.93 | 64.68 |
| 113k | Alpagasus | 84.91 | 81.00 | 53.66 | 39.90 | 64.87 |

Table 15: Ablation on selector architectures (CNN, MLP, Transformer) using 10k CPQS-selected Reasoning-DeepSeek samples. Metrics are reported on GSM8K, Math_500, HuanEval, and GPQA, with the simple average.

| Size | Method | GSM8K | Math_500 | HuanEval | GPQA | Average |
|------|--------|-------|----------|----------|------|---------|
| 10k | CNN | 85.37 | 72.40 | 67.68 | 36.36 | **65.45** |
| 10k | MLP | 87.04 | 72.80 | 60.37 | 29.29 | 62.38 |
| 10k | Transformer | 84.15 | 73.20 | 67.07 | 30.81 | 63.81 |

mum output token length set to **8000**). Table 16 reports results on GSM8K, Math_500, HuanEval, and GPQA, as well as the simple average.

As shown in Table 16, our method achieves the highest average performance among all compared baselines, indicating a stronger ability to identify samples that are most beneficial for the target model.

### A.7.2 COMPUTATIONAL COST AND THROUGHPUT ANALYSIS

To provide a transparent accounting of efficiency, we report GPU memory consumption and throughput (samples per second) under identical hardware constraints. All methods were executed on an **NVIDIA RTX PRO 6000** GPU. Throughput was computed over **146,224** samples using total wall-clock processing time. Results are summarized in Table 17. As shown in the table below, our method ranks just behind Alpagasus and Superfliting in terms of GPU memory usage, and second only to Superfliting in throughput.

Table 16: Comparison with recent data-selection methods on *Reasoning-DeepSeek* (10k selected samples) using **Qwen2.5-7B-Instruct**. All methods are evaluated under identical generation settings.

| Size | Method | GSM8K | Math_500 | HuanEval | GPQA | Average |
|------|--------|-------|----------|----------|------|---------|
| 10k | Self | 85.37 | 72.40 | **67.68** | **36.36** | **65.45** |
| 10k | DS2 | 84.15 | 72.80 | 62.80 | 29.80 | 62.39 |
| 10k | SelectIT | **85.67** | **73.60** | 59.76 | 31.82 | 62.71 |
| 10k | Deita | 82.56 | 76.00 | 62.80 | 30.81 | 63.04 |

Table 17: GPU memory usage and throughput under identical hardware (NVIDIA RTX PRO 6000). Throughput is measured in samples per second over 146,224 samples. Best values per column are in **bold**.

| Method | GPU Memory (GB) | Throughput (samples/s) |
|--------|-----------------|------------------------|
| Self | 18 | 4.78 |
| MoDs | 24 | 1.40 |
| Alpagasus | 0 | 1.45 |
| Superfliting | 2 | 34.82 |
| DS2 | 30 | 1.44 |
| Deita | 50 | 2.03 |
| SelectIT | 26 | 1.13 |

