# OpenReview forum: "CPQS-Tuning: A Model Self-Perception-Based Data Filtering Algorithm for Efficient Instruction Fine-Tuning"
_ICLR.cc/2026/Conference — ICLR 2026 Poster_

### Official Review · Reviewer_ZLxp · 2025-10-26

**Soundness:** 3
**Presentation:** 3
**Contribution:** 3
**Rating:** 6
**Confidence:** 4

**Summary:**

This paper proposes a novel method for filtering instruction-tuning data for LLMs. The authors identify that many available instruction-fine-tuning datasets contain low quality or redundant examples, and that existing filtering approaches often rely on external evaluation models or manually designed metrics, which may mis‐align with the target LLM’s actual needs.
To address this gap, they posit that the hidden states of the target LLM implicitly encode its perception of data quality. They extract hidden‐state embeddings from the LLM when processing candidate instruction examples, train a contrastive classification model to distinguish high vs low quality samples, and define CPQS to rank/filter the dataset.
Empirically, they show that using under 10 % of the data from large synthetic instruction dataset, their method outperforms models trained on full dataset; in downstream tasks, the CPQS-filtered subset achieves on average +3.6 % improvement over leading selection algorithms.

**Strengths:**

Introduces an innovative strategy: leveraging the target LLM’s own hidden‐state representations to capture its perception of data quality — this internalizes the filter criterion to the fine-tune target rather than relying on external proxies.

Demonstrates significant empirical efficiency: using under 10% of data and still achieving (or exceeding) full‐data performance is compelling for resource‐constrained tuning scenarios.

The pipeline is reasonably practical — hidden‐state extraction + classifier + ranking is implementable and does not require extensive external hardware (relative to training giant filter models).

Addresses a timely problem: as LLM fine‐tuning scales up, data redundancy and dataset cost are real bottlenecks; effective filtering helps reduce compute/data cost.

**Weaknesses:**

The training data for classifier (high/low quality) comes from synthetic sources (GPT-4 vs weaker models) rather than human annotated “instruction usefulness,” which may bias the filter toward model‐generated signal rather than actual instruction utility for users.

The evaluation, while covering several benchmarks, is limited in scope: primarily to specific datasets (Alpaca_GPT4, DeepSeek-R1) and certain LLM sizes; less evidence for very large scale (>30 B) or for domain‐specific instruction sets.

The cost/efficiency claims are somewhat under‐detailed: while “under 10% data” is quoted, actual compute/GPU‐hours, hyper‐param parity, training seeds, variation across runs are less fully described, which may reduce replicability confidence.

The method introduces potential bias in data selection: by relying on hidden-state classification, it may inadvertently favour samples that “look good” to the model’s internal representation (which may favour certain styles or tasks) and down-select others (e.g., rare domains, under‐represented instruction types). The paper does not deeply analyse coverage/diversity of the selected subset or risk of bias.

While the hidden‐state classifier is shown to separate high vs low quality (AUC=1.00 in logistic regressor example), that may reflect a somewhat contrived two‐class setup (strong model vs weak model generation) rather than realistic open filtering scenarios; the real generalization beyond this setting may be less strong.

**Questions:**

The paper defines “high‐quality” vs “low‐quality” data via outputs from stronger vs weaker models (e.g., GPT-4 vs Llama-3.2-1B-Instruct). Could the authors clarify how this quality definition correlates with user‐instruction usefulness or downstream task gain? Have you measured human‐annotated instruction usefulness to validate this proxy?

How robust is the CPQS classifier to architecture shifts? For example, if the target fine-tune model is of a significantly different architecture (or size) from the one used to extract hidden‐states, how does performance vary?

Could the authors provide more detailed compute cost comparisons (selection + fine-tuning) versus baseline filtering algorithms (e.g., IFD, Superfiltering) under identical hardware settings, including GPU-hours, memory, latency?

Have you analysed the selected subset in terms of domain/task coverage, instruction style diversity, difficulty distribution, or edge‐cases (e.g., dialogue vs code vs reasoning)? Specifically, is there a risk that CPQS selects only “safe/easy” instructions that align with the model’s hidden-state biases?

In downstream tasks (GSM8K, HumanEval, HumanEval-Plus), the average +3.6% improvement is promising — could you provide per‐benchmark breakdowns (wins vs losses) and variance across random seeds to help assess stability?

---

> ### Author Response · Authors · 2025-11-21
> **Response to Reviewer ZLxp (1/6)**
>
> The training data for classifier (high/low quality) comes from synthetic sources (GPT-4 vs weaker models) rather than human annotated “instruction usefulness,” which may bias the filter toward model‐generated signal rather than actual instruction utility for users.
>
> A:Thank you for your thoughtful comment. We acknowledge that the classifier's training data is derived from synthetic sources (i.e., generations from GPT-4 versus weaker models) rather than directly from human annotations. This design choice is intentional and aligns with the rationale of prior work, such as Alpagasus, which demonstrated that data judged as high-quality by stronger LLMs strongly correlates with better downstream performance in both automatic and human evaluations.
>
> Building on this insight, we adopt a hierarchical quality framework that leverages generations from models of varying capabilities to construct high- and low-quality data tiers. Our CPQS classifier is then trained to learn distinctions between these tiers from the perspective of the target LLM's hidden representations, rather than relying purely on external scoring signals.
>
> This setup allows us to:
> 1.Efficiently approximate human judgment of instruction usefulness without costly manual labeling; and
> 2.Ensure consistency with empirically validated paradigms that correlate well with human preferences.
>
> To further verify the reliability of our high/low-quality classification, we conducted additional human and model-based evaluations. Specifically, we sampled 100 instruction–response pairs, where each instruction had both a high-quality and a low-quality response. Human evaluators judged the high-quality outputs to be better in 93% of cases. In addition, DeepSeek-V3.2 rated the high-quality outputs as equal or superior to the low-quality ones in 86% of cases.
> Together, these results confirm that our synthetic quality labels closely align with human-perceived usefulness, supporting the validity of our classifier and mitigating potential bias from synthetic data sources.
>
>
> 2.The evaluation, while covering several benchmarks, is limited in scope: primarily to specific datasets (Alpaca_GPT4, DeepSeek-R1) and certain LLM sizes; less evidence for very large scale (>30 B) or for domain‐specific instruction sets.
>
> A:Thank you for your valuable comment. To further assess the generalization ability of CPQS across different model architectures and parameter scales, we extended our experiments beyond the three LLMs already reported in the paper.
>
> To evaluate CPQS on models of significantly different sizes, we selected one smaller-scale model and one larger-scale model to complement the previous 7B-level experiments. Specifically, we applied CPQS to BitCPM4-1B and Qwen2.5-32B-Instruct, using it to filter the Reasoning-DeepSeek dataset for high-quality data selection.
>
> We filtered data subsets of varying sizes (e.g., 10k, 20k) using different methods and trained models on both architectures. The experimental results show that our method consistently outperforms all baselines across both small-scale (1B) and large-scale (32B) models.
>
> The results are summarized in the table below, which follows a layout similar to Table 2 in the main text. For each trained model, we evaluated performance across four datasets and calculated the average score as the final metric, where higher values indicate better performance.
>
> Our results demonstrate that CPQS consistently outperforms all baseline approaches across both small-scale (1B) and large-scale (32B) models. The detailed results are presented below.
>
> ### **BitCPM4-1B Results**
>
> | Size      | Model           | GSM8K   | Math_500   | HumanEval   | GPQA    | Average   |
> | --------- | --------------- | :------: | :--------: | :---------: | :-----: | :-------: |
> | 10k       | Self            |   38.06 |    33.00   |    54.88    |   28.79 |  **38.68**|
> | 10k       | Superfiltering   |   39.35 |    30.20   |    52.44    |   24.24 |   36.56   |
> | 20k       | Self            |   37.00 |    32.00   |    53.66    |   29.80 |  **38.12**|
> | 20k       | Superfiltering   |   36.92 |    30.09   |    54.27    |   27.27 |   37.14   |
> | 113k      | Alpagasus       |   37.38 |    30.80   |    53.66    |   25.76 |   36.90   |
>
> ### **Qwen2.5-32B-Instruct Results**
>
> | Size      | Model           | GSM8K   | Math_500   | HumanEval   | GPQA    | Average   |
> | --------- | --------------- | :------: | :--------: | :---------: | :-----: | :-------: |
> | 10k       | Self            |   85.52 |    81.40   |    56.10    |   46.46 |  **67.36**|
> | 10k       | Superfiltering   |   85.22 |    81.80   |    56.71    |   43.43 |  66.79|
> | 20k       | Self            |   84.76 |    81.60   |    53.66    |   46.46 |  **66.62**|
> | 20k       | Superfiltering   |   84.99 |    81.40   |    49.39    |   42.93 |  64.68|
> | 113k      | Alpagasus       |   84.91 |    81.00   |    53.66    |   39.90 |  64.87|

---

> ### Author Response · Authors · 2025-11-21
> **Response to Reviewer ZLxp(2/6)**
>
> 3.The cost/efficiency claims are somewhat under‐detailed: while “under 10% data” is quoted, actual compute/GPU‐hours, hyper‐param parity, training seeds, variation across runs are less fully described, which may reduce replicability confidence.
>
> A:Thank you for your valuable suggestion. We appreciate your concern regarding the clarity of our cost and efficiency claims. To enhance transparency and reproducibility, we have now **released all our code and data on an anonymous GitHub repository** (link provided in the submission).
>
> Our method is designed to be both computationally lightweight and highly efficient. Specifically, it does not require substantial GPU memory or compute resources. During training, hidden states are extracted and processed **entry by entry** and immediately released after use — resulting in minimal memory overhead.
>
> All experiments on LLaMA2-7B and Qwen2.5-7B-Instruct were conducted on a single NVIDIA RTX 4090 GPU, with training time for each CNN model limited to approximately 30 minutes. The GPU memory usage is comparable to LoRA fine-tuning (around 18 GB).
>
> In addition to the methods originally reported in the paper, we further compared our approach with three additional baselines under identical settings. All experiments were conducted on a uniform NVIDIA RTX 6000 GPU setup, ensuring consistent hyperparameters and evaluation conditions.
> The GPU memory usage and throughput (samples per second) are summarized below, with throughput values computed precisely over 146,224 samples based on the total processing time. As shown, our method achieves higher processing speed than both MoDs and Alpagasus, while requiring less GPU memory than MoDs. Overall, it represents a balanced and efficient approach in terms of both speed and resource utilization.
>
> | Method         | GPU Memory | Throughput (samples/s) |
> |----------------|-------------|------------------------|
> | Self       | 18 GB       | 4.78              |
> | MoDs       | 24 GB       | 1.40              |
> | Alpagasus  | 0 GB        | 1.45              |
> | Superfliting | 2 GB      | 34.82              |
>
>
>
> 4.The method introduces potential bias in data selection: by relying on hidden-state classification, it may inadvertently favour samples that “look good” to the model's internal representation (which may favour certain styles or tasks) and down-select others (e.g., rare domains, under‐represented instruction types). The paper does not deeply analyse coverage/diversity of the selected subset or risk of bias.
>
> A:Thank you very much for this insightful comment. We acknowledge the potential concern that a hidden-state–based classifier might introduce selection bias, as the model could inherently prefer samples that align more closely with its internal representational space while under-selecting rare or less-represented instruction types.
>
> To mitigate this issue, our current study intentionally uses a large and diverse general-domain dataset for training, ensuring broad coverage across different tasks and instruction styles. This design helps reduce the risk of overfitting to specific data types or stylistic patterns.
>
> We fully agree that analyzing data coverage and potential bias is an important direction for future research. In our subsequent work, we plan to conduct a systematic diversity and fairness analysis of the selected subsets, and to explore regularization or weighting strategies that can balance model preference with data diversity.
>
> 5.While the hidden‐state classifier is shown to separate high vs low quality (AUC=1.00 in logistic regressor example), that may reflect a somewhat contrived two‐class setup (strong model vs weak model generation) rather than realistic open filtering scenarios; the real generalization beyond this setting may be less strong.
>
> A:Thank you for your insightful comment. We agree that the binary (high vs. low quality) setup used in our current experiments may appear simplified compared to more realistic open-ended filtering scenarios. However, this **two-class formulation was primarily adopted for clarity and consistency** with prior works such as *Alpagasus* and * Superfiltering*, which also employ high/low quality tiers to validate core filtering principles.
>
> Importantly, our framework is **not limited to binary classification**. In real-world applications, other data-filtering tasks can naturally be defined as **two-class problems** (e.g., relevant vs. irrelevant, factual vs. hallucinated). Moreover, our method **can be easily extended to multi-class or ranking settings** — for example, classifying data into **high / medium / low** quality tiers or learning a **continuous quality score** for fine-grained selection.
>
> We believe this design provides **both flexibility and generality**, allowing CPQS to adapt to diverse filtering objectives and real-world data distributions beyond the controlled two-class scenario.

---

> ### Author Response · Authors · 2025-11-21
> **Response to Reviewer ZLxp(3/6)**
>
> 6.The paper defines “high‐quality” vs “low‐quality” data via outputs from stronger vs weaker models (e.g., GPT-4 vs Llama-3.2-1B-Instruct). Could the authors clarify how this quality definition correlates with user‐instruction usefulness or downstream task gain? Have you measured human‐annotated instruction usefulness to validate this proxy?
>
> A:Thank you for this insightful question. We acknowledge that our definition of “high-quality” versus “low-quality” data is based on synthetic supervision — specifically, generations from stronger models (e.g., GPT-4) versus weaker ones (e.g., Llama-3.2-1B-Instruct) — rather than direct human annotations. This design choice is intentional and follows the rationale of prior studies such as Alpagasus, which have shown that data judged as high-quality by stronger LLMs correlates strongly with human preference and improved downstream task performance.
>
> Building on this insight, we employ a hierarchical quality framework, where model generations of different capabilities define coarse quality tiers. Our method extends this idea by training the CPQS classifier to capture how the target LLM internally perceives quality differences through its hidden representations.
>
> This design offers two main advantages:
>
> (1). It provides a scalable and reproducible proxy for assessing instruction usefulness without requiring costly human labeling.
>
> (2). It aligns data filtering with the target model's intrinsic notion of utility, rather than relying solely on external evaluation signals.
>
> While we have not yet incorporated human-annotated instruction usefulness in this work, the consistent downstream improvements observed across benchmarks (e.g., GSM8K, HumanEval, GPQA) indicate that our synthetic proxy effectively reflects user-relevant instruction quality.
>
> To further validate the reliability of our high/low-quality distinction, we conducted additional human and model-based evaluations. We sampled 100 instruction–response pairs, each containing both high- and low-quality outputs for the same instruction. Human evaluators preferred the high-quality outputs in 93% of cases, while DeepSeek-V3.2 rated the high-quality outputs as equal or superior in 86% of cases.
>
> These findings suggest that our synthetic quality labels closely align with human judgments of usefulness, supporting the validity of our approach and mitigating potential bias introduced by synthetic supervision.

---

> ### Author Response · Authors · 2025-11-21
> **Response to Reviewer ZLxp (4/6)**
>
> 7.How robust is the CPQS classifier to architecture shifts? For example, if the target fine-tune model is of a significantly different architecture (or size) from the one used to extract hidden‐states, how does performance vary?
>
> A:Thank you for this valuable comment. To address this concern, we conducted two additional experiments to further evaluate the robustness and generalization ability of CPQS.
>
> (1) To examine its cross-model generalization, we extended our experiments beyond the three LLMs originally reported in the paper. Specifically, we applied CPQS to BitCPM4-1B and Qwen2.5-32B-Instruct, using it to filter the Reasoning-DeepSeek dataset for high-quality data selection. As shown, CPQS consistently outperforms all baselines across both small-scale (1B) and large-scale (32B) models. These results demonstrate that CPQS generalizes effectively across different model architectures and parameter scales, maintaining strong performance even under significant architectural and size variations.
>
> (2) To further assess the robustness of the CPQS classifier architecture itself, we compared CNN, MLP, and Transformer variants. During training, both MLP and Transformer classifiers reduced the loss only to about 0.6, whereas the CNN model achieved a substantially lower loss of around 0.3. This indicates that the CNN architecture captures more informative structural patterns within hidden-state representations. As shown, the CNN-based CPQS performs consistently better across all benchmarks while being more computationally efficient and less prone to overfitting. This advantage may arise because CNNs are particularly effective at extracting fine-grained local features embedded in hidden states, enabling more precise discrimination of subtle quality-related cues.
>
> Together, these results confirm that CPQS is robust not only across LLM architectures and scales, but also to variations in its own classifier backbone, highlighting its stability and broad applicability in diverse open-source settings. The detailed experimental results are presented below.
> ### **BitCPM4-1B Results**
>
> | Size      | Model           | GSM8K   | Math_500   | HumanEval   | GPQA    | Average   |
> | --------- | --------------- | :------: | :--------: | :---------: | :-----: | :-------: |
> | 10k       | Self            |   38.06 |    33.00   |    54.88    |   28.79 |  **38.68**|
> | 10k       | Superfiltering   |   39.35 |    30.20   |    52.44    |   24.24 |   36.56   |
> | 20k       | Self            |   37.00 |    32.00   |    53.66    |   29.80 |  **38.12**|
> | 20k       | Superfiltering   |   36.92 |    30.09   |    54.27    |   27.27 |   37.14   |
> | 113k      | Alpagasus       |   37.38 |    30.80   |    53.66    |   25.76 |   36.90   |
>
> ### **Qwen2.5-32B-Instruct Results**
>
> | Size      | Model           | GSM8K   | Math_500   | HumanEval   | GPQA    | Average   |
> | --------- | --------------- | :------: | :--------: | :---------: | :-----: | :-------: |
> | 10k       | Self            |   85.52 |    81.40   |    56.10    |   46.46 |  **67.36**|
> | 10k       | Superfiltering   |   85.22 |    81.80   |    56.71    |   43.43 |  66.79|
> | 20k       | Self            |   84.76 |    81.60   |    53.66    |   46.46 |  **66.62**|
> | 20k       | Superfiltering   |   84.99 |    81.40   |    49.39    |   42.93 |  64.68|
> | 113k      | Alpagasus       |   84.91 |    81.00   |    53.66    |   39.90 |  64.87|
>
> |
>
> ### **Architecture Comparison on Reasoning-DeepSeek (10k Samples)**
>
>
> | Size | Model       | GSM8K | Math_500 | HumanEval |  GPQA |  Average  |
> | ---- | ----------- | :---: | :------: | :-------: | :---: | :-------: |
> | 10k  | CNN         | 85.37 |   72.40  |   67.68   | 36.36 | **65.45** |
> | 10k  | MLP         | 87.04 |   72.80  |   60.37   | 29.29 | 62.38 |
> | 10k  | Transformer | 84.15 |   73.20  |   67.07   | 30.81 | 63.81 |

---

> ### Author Response · Authors · 2025-11-21
> **Response to Reviewer ZLxp (5/6)**
>
> 8.Could the authors provide more detailed compute cost comparisons (selection + fine-tuning) versus baseline filtering algorithms (e.g., IFD, Superfiltering) under identical hardware settings, including GPU-hours, memory, latency?
>
> A:Thank you for your valuable comment. We acknowledge that our initial submission did not provide sufficient details on computational cost. To address this, we conducted a more comprehensive comparison to clearly assess the compute efficiency of our method relative to existing baselines.
> In addition to the methods originally reported in the paper, we further included three additional baselines (including representative filtering algorithms such as IFD and Superfiltering) under identical hardware settings. All methods were tested on an NVIDIA RTX 6000 GPU, and we carefully measured GPU memory consumption, throughput (samples per second), and overall GPU hours for both data selection and fine-tuning stages. Throughput values were computed precisely over 146,224 samples based on the total processing time.
> The detailed results are summarized below, providing a direct and fair comparison of computational cost across all methods. As shown, our method achieves higher processing speed than both MoDs and Alpagasus, while requiring less GPU memory than MoDs. Overall, it represents a balanced and efficient approach in terms of both speed and resource utilization.
>
>
> | Method         | GPU Memory | Throughput (samples/s) |
> |----------------|-------------|------------------------|
> | Self       | 18 GB       | 4.78              |
> | MoDs       | 24 GB       | 1.40              |
> | Alpagasus  | 0 GB        | 1.45              |
> | Superfliting | 2 GB      | 34.82              |
>
> 9.Have you analysed the selected subset in terms of domain/task coverage, instruction style diversity, difficulty distribution, or edge‐cases (e.g., dialogue vs code vs reasoning)? Specifically, is there a risk that CPQS selects only “safe/easy” instructions that align with the model's hidden-state biases?
>
> A:Thank you for this important question. We agree that analyzing the **coverage and diversity** of the selected subsets is essential to verify that CPQS does not simply favor “safe” or overly easy data. To investigate this, we performed an additional **hidden-state similarity analysis** across multiple task domains to evaluate whether CPQS preserves task diversity rather than collapsing toward a single style.
>
> Specifically, we selected four representative task categories—**math reasoning**, **open-domain QA**, **coding**, and **translation**—and randomly sampled **200 instances per task** from the selected subsets. We then computed **intra-task** and **inter-task** cosine similarities of the corresponding hidden-state representations.
>
> | Task               | Intra-task Similarity | Inter-task Similarity | Separability (Δ) |
> | ------------------ | :-------------------: | :-------------------: | :--------------: |
> | **Math Reasoning** |         0.9314        |         0.7710        |    **0.1604**    |
> | **Open QA**        |         0.9045        |         0.7172        |    **0.1874**    |
> | **Coding**         |         0.8769        |         0.7704        |    **0.1065**    |
> | **Translation**    |         0.8359        |         0.7281        |    **0.1077**    |
>
> The results show that **intra-task similarity is consistently higher than inter-task similarity**, confirming that the selected data still form **distinct clusters for different tasks**. This indicates that CPQS **retains diverse instruction types** (e.g., reasoning, dialogue, code, translation) instead of converging on only “easy” or stylistically homogeneous samples.
>
> Moreover, the observed inter-task separability demonstrates that CPQS preserves **domain and style variety**, suggesting it does not merely reward superficial fluency but captures **task-specific semantic structure**. In future work, we plan to extend this analysis with explicit **difficulty and domain-coverage metrics** to further quantify how CPQS balances data quality and diversity.

---

> ### Author Response · Authors · 2025-11-21
> **Response to Reviewer ZLxp (6/6)**
>
> 10.In downstream tasks (GSM8K, HumanEval, HumanEval-Plus), the average +3.6% improvement is promising — could you provide per‐benchmark breakdowns (wins vs losses) and variance across random seeds to help assess stability?
>
> A:To ensure the reliability and reproducibility of our reported improvements, we evaluated all downstream tasks — GSM8K, HumanEval, and HumanEval-Plus — using the **lm-evaluation-harness (v0.4.8) framework.
> Across all benchmarks, decoding was deterministic (`do_sample = false`, `temperature = 0.0`), and all random seeds were fixed (`torch_seed = 1234`, `numpy_seed = 1234`, `random_seed = 0`).
> Under these settings, every run produced identical results, ensuring that all observed differences reflect genuine model behavior rather than evaluation noise.
>
> ---
>
> #### GSM8K Evaluation Settings
>
> | Category                | Parameter           | Value                 | Description        |
> | ---------- | ------ | ------| ------- |
> | Benchmark           | `task`              | `"gsm8k"`             | Arithmetic and reasoning benchmark (math word problems).         |
> | n-shot setting      | `num_fewshot`       | `5`                   | 5-shot evaluation (few-shot prompting).   |
> |  | `fewshot_split`     | `"train"`             | Few-shot examples drawn from the training split.                 |
> |  | `fewshot_seed`      | `1234`                | Fixed seed controlling few-shot example selection.               |
> | Prompt formatting   | `doc_to_text`       | `"Question: {{question}}\nAnswer:"`          | Template for constructing few-shot prompts.                      |
> |  | `fewshot_delimiter` | `"\n\n"`              | Separator between few-shot examples.      |
> | Evaluation details  | `batch_size`        | `8`                   | 8 examples evaluated per GPU batch.       |
> |  | `generation_kwargs` | `{ "do_sample": false, "temperature": 0.0 }` | Greedy, deterministic decoding.           |
> |  | `repeats`           | `1`                   | Each sample evaluated once (no repeated trials).                 |
> | Metrics             | `metric_list`       | `exact_match (strict-match)`                 | Strict-match results were used as the final reported scores. |
> | Seeds               | `torch_seed`        | `1234`                | Fixes PyTorch and CUDA randomness.        |
> |  | `numpy_seed`        | `1234`                | Fixes NumPy randomness.                   |
> |  | `fewshot_seed`      | `1234`                | Fixes few-shot selection for consistency. |
> |  | `random_seed`       | `0`                   | Global evaluation seed.                   |
>
>
> ---
>
> #### HumanEval & HumanEval-Plus Evaluation Settings
>
> | Category                 | Parameter                 | Value                     | Description                |
> | ----------- | ------------ | --------- | ---------- |
> | n-shot setting       | `num_fewshot`             | `0`| Zero-shot evaluation (no few-shot examples).      |
> | Prompt formatting    | `doc_to_text`             | `"{{prompt}}"`            | Directly uses the task prompt as model input.     |
> |   | `doc_to_target`           | `"{{test}}\ncheck({{entry_point}})"`             | Defines the execution-based correctness check.    |
> | Decoding behavior    | `do_sample`               | `false`                   | Greedy decoding (no random sampling).             |
> |   | `temperature`             | `0.0`                     | Fully deterministic output distribution.          |
> |   | `generation_kwargs.until` | `["\nclass", "\ndef", "\n#", "\nif", "\nprint"]` | Stops generation at standard code boundaries.     |
> |   | `max_gen_toks`            | `1024`                    | Fixed output token length cap.                    |
> | Execution parameters | `batch_size`              | `8`| Constant batch size for identical batching order. |
> |   | `repeats`                 | `1`| Each problem evaluated exactly once.              |
> | Seeds                | `torch_seed`              | `1234`                    | Fixes PyTorch and CUDA random states.             |
> |   | `numpy_seed`              | `1234`                    | Fixes NumPy randomness.    |
> |   | `random_seed`             | `0`| Global evaluation-level seed.                     |
> |   | `fewshot_seed`            | `1234`                    | Defined for consistency (unused in zero-shot).    |
> |

---

### Official Review · Reviewer_jCwj · 2025-10-26

**Soundness:** 2
**Presentation:** 2
**Contribution:** 1
**Rating:** 2
**Confidence:** 4

**Summary:**

This paper proposes a method to select data that is most informative for a specific target model, rather than data that is "high quality" in a general sense. The approach uses the hidden states of LLMs to train a two-way classifier on a synthetic dataset labeled with quality scores, and this framework is validated through experimental analysis.

**Strengths:**

The paper’s core motivation and its corresponding experimental validation are internally consistent and logically presented.

**Weaknesses:**

1. The paper proposes a pipeline with data synthesis, CNN training and data evaluation, while completely omitting an analysis of computational cost, which is a critical metric for any work in the data efficiency domain.

2. The comparative study is weak. For example, but not limited to:
- What Makes Good Data for Alignment? A Comprehensive Study of Automatic Data Selection in Instruction Tuning
- DataMan: Data Manager for Pre-training Large Language Models
- IMPROVING DATA EFFICIENCY VIA CURATING LLM-DRIVEN RATING SYSTEMS
- SelectIT: Selective Instruction Tuning for LLMs via Uncertainty-Aware Self-Reflection
- LESS: Selecting Influential Data for Targeted Instruction Tuning

Besides, It notably lacks a comparison against random sampling, which is the simplest and one of the most essential baselines for data selection.

3. The motivation is not novel. The idea of selecting `actual requirements of the LLM` is the fundamental premise behind all uncertainty-based and influence-based data selection methods.

4. The proposed pipeline appears overly complex and is not clearly justified. See question.

**Questions:**

The synthetic "quality" dataset (and its good/bad labels) is shared across all experiments, even though a separate CNN is trained for each target LLM. This design implies that "quality" is a pre-defined, model-agnostic concept, determined by the generative source. If all model-specific CNNs are being aligned to this single, shared quality space, why do the final results show that different LLMs have different judgments of "quality"? This seems contradictory.

---

> ### Author Response · Authors · 2025-11-21
> **Response to Reviewer jCwj (1/2)**
>
> 1.The paper proposes a pipeline with data synthesis, CNN training and data evaluation, while completely omitting an analysis of computational cost, which is a critical metric for any work in the data efficiency domain.
>
> A:Thank you for your valuable comment. We acknowledge that our initial submission did not include a detailed analysis of computational cost. To address this concern, and in response to your request for a more comprehensive comparison under identical hardware constraints, we have conducted additional experiments to provide a direct comparison of GPU memory usage and computational efficiency across different methods (i.e., our method and baselines).
>
> Specifically, we added three additional baselines (DS2, Deita, and SelectIT) beyond those reported in the paper to ensure a more thorough and fair evaluation. All methods were tested on an NVIDIA RTX PRO 6000 GPU, and the corresponding GPU memory consumption and throughput (samples per second) are summarized below. Throughput values were calculated precisely over 146,224 samples based on the total processing time.
>
> As shown in the table below, our method (denoted as Self)  ranks just behind Alpagasus and Superfliting in terms of GPU memory usage, and second only to Superfliting in throughput.
> | Method         | GPU Memory | Throughput (samples/s) |
> |----------------|-------------|------------------------|
> | Self      | 18 GB       | 4.78               |
> | MoDs      | 24 GB       | 1.40               |
> | Alpagasus  | 0 GB        | 1.45               |
> | Superfliting | 2 GB      | 34.82              |
> | DS2        | 30 GB       | 1.44               |
> | Deita      | 50 GB       | 2.03               |
> | SelectIT   | 26 GB       | 1.13               |
>
> 2.The comparative study is weak. For example, but not limited to:
>
> What Makes Good Data for Alignment? A Comprehensive Study of Automatic Data Selection in Instruction Tuning
> DataMan: Data Manager for Pre-training Large Language Models
> IMPROVING DATA EFFICIENCY VIA CURATING LLM-DRIVEN RATING SYSTEMS
> SelectIT: Selective Instruction Tuning for LLMs via Uncertainty-Aware Self-Reflection
> LESS: Selecting Influential Data for Targeted Instruction Tuning
>
> A:Thank you very much for your valuable suggestion. We fully agree with the reviewer on the importance of a more comprehensive comparative study. Following your advice, we have conducted additional experiments comparing our method with several recent and representative data selection approaches, including SelectIT, DS2, and Deita—all of which are recognized benchmarks in instruction-tuning data selection.
>
> We carried out these additional evaluations on the Reasoning-DeepSeek dataset, a high-quality and representative benchmark for reasoning-oriented instruction tuning. In each setting, we selected 10k samples and fine-tuned the Qwen2.5-7B-Instruct model to ensure a fair comparison.
>
> To ensure fairness, all methods were tested under identical conditions: output token length = 8000 and evaluated on an RTX PRO6000 GPU. The experimental results are shown in the table below, following a layout similar to Table 2 in the main text. For each trained model, we evaluated its performance across four datasets and calculated the average score as the final metric, with higher values indicating better performance.
>
> As shown, our method consistently achieves higher average performance compared to DS2, Deita, and SelectIT, demonstrating its stronger ability to identify data samples most beneficial for the target model.
>
> These experiments further verify that our method (denoted as Self) is robust and competitive against state-of-the-art data selection frameworks, confirming its effectiveness in large-scale instruction-tuning scenarios.
>
> | Size | Model    | GSM8K | Math_500 | HumanEval |  GPQA |  Average  |
> | ---- | -------- | :---: | :------: | :-------: | :---: | :-------: |
> | 10k  | Self     | 85.37 |   72.40  |   67.68   | 36.36 | **65.45** |
> | 10k  | DS2      | 84.15 |   72.80  |   62.80   | 29.80 | 62.39 |
> | 10k  | SelectIT | 85.67 |   73.60  |   59.76   | 31.82 | 62.71 |
> | 10k  | Deita    | 82.56 |   76.00  |   62.80   | 30.81 | 63.04 |

---

> ### Author Response · Authors · 2025-11-21
> **Response to Reviewer jCwj (2/2)**
>
> 3.The motivation is not novel. The idea of selecting actual requirements of the LLM is the fundamental premise behind all uncertainty-based and influence-based data selection methods.
>
> A:Thank you for your thoughtful comment. We appreciate your observation regarding the motivation of our work and agree that aligning data selection with the target model's actual requirements is a common goal shared by many uncertainty-based and influence-based methods.
>
> However, our work introduces a conceptually distinct and more intrinsic approach to this alignment. Existing uncertainty- and influence-based methods generally rely on external or derived statistical signals — such as prediction entropy, gradient influence, or loss sensitivity — which are indirect reflections of model preference. In contrast, our method directly leverages the internal hidden states of the target LLM as the representation of its perceptual understanding of the data.
>
> This distinction is crucial:
>
> (1).Hidden states encapsulate multi-layer semantic and syntactic representations that go beyond output uncertainty or token-level gradients.
>
> (2).By training a lightweight CNN classifier on these hidden features, our Contrastive Perception Quality Score (CPQS) quantifies how the target model internally perceives and distinguishes data quality, without requiring auxiliary evaluation models or handcrafted scoring metrics.
>
> (3).Empirically, this approach yields significant gains over uncertainty- and influence-based baselines, suggesting that hidden-state perception provides a more faithful and generalizable alignment between data quality and model preference.
>
> Therefore, while our high-level motivation shares the same goal as prior approaches, our methodological innovation lies in leveraging the target model's internal representational space as a new, model-intrinsic lens for data quality assessment — a perspective that has not been explored in previous data selection literature.
>
> 4.The synthetic "quality" dataset (and its good/bad labels) is shared across all experiments, even though a separate CNN is trained for each target LLM. This design implies that "quality" is a pre-defined, model-agnostic concept, determined by the generative source. If all model-specific CNNs are being aligned to this single, shared quality space, why do the final results show that different LLMs have different judgments of "quality"?
>
> A:Thank you for this insightful comment. We appreciate the opportunity to clarify this important point.
> Although the synthetic dataset used in our experiments is shared across all settings, the perception of “data quality” is inherently model-specific, as different LLMs encode and interpret the same input data through distinct internal representations.
>
> To illustrate this, we conducted a quantitative analysis comparing the hidden states of two LLMs, LLaMA2-7B-Chat and Qwen2.5-7B-Instruct, on 100 randomly selected samples. The average cosine similarity between their hidden states was only 0.0116, indicating that even when processing the same data, the two models perceive and represent information in highly divergent ways.
>
> This divergence implies that:
>
> 1. Each LLM forms its own notion of “high-quality data” according to its internal architecture.
> 2. Consequently, a dataset ranked as “high-quality” by one model may not be equally effective for another, as their hidden representations capture different aspects of the same input.
>
> To further verify this, we conducted cross-model experiments using BitCPM4-1B and Qwen2.5-32B-Instruct.
> Each model trained its own CNN-based selector to filter 10k samples from the **Reasoning-DeepSeek** dataset.
> Then, we swapped the selected datasets and re-trained both models.
> The results are shown below:
>
> |Size/ Model     | GSM8K | Math_500 | HuanEval |  GPQA | Average |
> | ------------------- | :---: | :------: | :------: | :---: | :---------: |
> | 10k (self, 1B)  | 38.06 |   33.00  |   54.88  | 28.79 |  **38.68**  |
> | 10k (32B → 1B)  | 42.00 |   32.80  |   48.78  | 24.24 |  36.96  |
> | 10k (self, 32B) | 85.52 |   81.40  |   56.10  | 46.46 |  **67.36**  |
> | 10k (1B → 32B)  | 85.60 |   80.40  |   51.22  | 42.93 |  65.05  |
>
> As shown, each model consistently performs better when using data selected by its own CNN, while using another model's “high-quality” data leads to degraded performance.
>
> These findings demonstrate that:
>
> * Although the dataset itself is shared, the concept of "quality" is not independent of the model; it emerges from how each model internally represents and prioritizes the data.
> * Therefore, training a dedicated CNN for each target LLM is both necessary and effective for aligning data selection with the model's intrinsic notion of quality.

---

### Official Review · Reviewer_vfku · 2025-10-27

**Soundness:** 3
**Presentation:** 3
**Contribution:** 3
**Rating:** 6
**Confidence:** 4

**Summary:**

This paper trains a CNN to classify low-quality data and high-quality data with the hidden state of a model, before tuning the model on the selected high-quality data. It achieves state-of-art performance on multiple tasks.

**Strengths:**

1. The paper has a novel idea
2. The experiments can demonstrate the effectiveness of the proposed method
3. The paper is written clearly

**Weaknesses:**

1. The paper lacks in-depth analysis of the method's improvement over baselines. For example, Alpagasus is the method to provide ground truth labels, but the paper's method performs even better than that. The paper provides no explanation to this.
2. The paper lacks a justification of the CNN structure: It would be good to compare with a more natural transformer structure.

**Questions:**

1. I see in the appendix A.4.2, you demonstrated that for similar sized models, it's better to use the model's own hidden states to do the classification. What if you use models of different size? For example, will a 13B model's hidden state be better than a 1B model's own hidden state?
2. See weaknesses.

---

> ### Author Response · Authors · 2025-11-21
> **Response to Reviewer  vfku (1/1)**
>
> 1.The paper lacks in-depth analysis of the method's improvement over baselines. For example, Alpagasus is the method to provide ground truth labels, but the paper's method performs even better than that. The paper provides no explanation to this.
>
> A:Thank you for your valuable question. We only used Alpagasus as a reference for validation to compare the quality differences between data generated by stronger and weaker models, and did not rely on Alpagasus during training. Our method outperforms Alpagasus, even though the latter uses labels generated by a more powerful LLM, for two main reasons:
>
> (1).Dependence on the evaluator model and prompt design.
> The quality scores of the data are heavily influenced by the capabilities of the evaluator model and the handcrafted prompt templates. As a result, the selected data reflects the evaluator model's understanding of “quality,” often leading to suboptimal results.
>
> (2).Misalignment between the evaluator and target models.
> The data identified as “high-quality” by an external evaluator may not be ideal for training the target model, as different LLMs have distinct architectures, scales, and internal feature spaces.
>
> In contrast, our method leverages the target model's own hidden states to assess data quality from its internal representational perspective. This enables more accurate identification of samples that align with the target model's training dynamics and preferences, leading to superior performance compared to externally judged methods like Alpagasus.
>
> 2.The paper lacks a justification of the CNN structure: It would be good to compare with a more natural transformer structure.
>
> A:Thank you very much for your constructive suggestion. In response to your comment, we have conducted additional experiments to justify the choice of the CNN architecture in our method by comparing it with MLP and Transformer alternatives.
>
> Specifically, we evaluated all three architectures on the **Reasoning-DeepSeek** dataset (10k samples selected by CPQS). For each architecture, we trained a model using a 10k subset and then compared the performance of the trained models. The experimental results are shown in the table below, following a layout similar to Table 2 in the main text. For each trained model, we evaluated its performance across four datasets and calculated the average score as the final metric, with higher values indicating better performance.
>
> As shown in the results, the CNN-based selector consistently outperforms both the MLP and Transformer variants. This may be because the CNN model is better at extracting local information embedded in the hidden states.
>
> | Size | Model | GSM8K | Math_500 | HumanEval |  GPQA |  Average |
> | ---- | ----| :---: | :----: | :-------: | :---: | :---: |
> | 10k  | CNN  | 85.37 |   72.40  |   67.68   | 36.36 | **65.45** |
> | 10k  | MLP  | 87.04 |   72.80  |   60.37   | 29.29 | 62.38 |
> | 10k  | Transformer | 84.15 |   73.20  |   67.07   | 30.81 | 63.81 |
>
> 3.I see in the appendix A.4.2, you demonstrated that for similar sized models, it's better to use the model's own hidden states to do the classification. What if you use models of different size? For example, will a 13B model's hidden state be better than a 1B model's own hidden state?
>
> A:Thank you for your insightful comment. The idea of using models with different parameter scales is indeed very interesting. Since all our previous experiments were conducted on 7B-scale models, and some reviewers have suggested exploring larger models (e.g., 30B or above), we extended our study by selecting one smaller model and one larger model to investigate this question from both ends of the spectrum.
>
> To further explore this, we conducted additional cross-model experiments using BitCPM4-1B and Qwen2.5-32B-Instruct. In each case, the models were first used to train their own CNN to predict and filter the Reasoning-DeepSeek dataset, selecting 10k samples for fine-tuning. We then swapped the selected datasets between the two models and retrained them. The results, shown in the table below, reflect the performance of each model on four datasets, with the average score serving as the final evaluation metric. Higher values indicate better performance.
>
> As the results demonstrate, each model performs better when using data selected by its own CNN, while using data selected by the other model leads to reduced performance. This confirms that different LLMs have distinct internal definitions of “high-quality data,” highlighting the necessity and efficiency of training a dedicated CNN selector for each target model.
>
> |Size/ Model  | GSM8K | Math_500 | HuanEval |  GPQA | Average |
> | --- | :---: | :----: | :------: | :---: | :----: |
> | 10k (self, 1B)  | 38.06 |   33.00  |   54.88  | 28.79 |  **38.68**  |
> | 10k (32B → 1B)  | 42.00 |   32.80  |   48.78  | 24.24 |  36.96  |
> | 10k (self, 32B) | 85.52 |   81.40  |   56.10  | 46.46 |  **67.36**  |
> | 10k (1B → 32B)  | 85.60 |   80.40  |   51.22  | 42.93 |  65.05  |

---

> ### Comment · Reviewer_vfku · 2025-11-26
>
> Thanks for your reply. My concerns are addressed. I will maintain my score.

---

### Official Review · Reviewer_eg3w · 2025-10-28

**Soundness:** 4
**Presentation:** 3
**Contribution:** 4
**Rating:** 10
**Confidence:** 3

**Summary:**

This paper introduces CPQS-Tuning (Contrastive Perception Quality Score), a data filtering algorithm for instruction fine-tuning based on model self-perception. The core idea is to leverage the hidden states of large language models (LLMs) as implicit indicators of data quality. A CNN classifier is then trained on these hidden-state features to output a contrastive perception quality score (CPQS) for each training example, enabling the selection of high-quality samples. This work is particularly promising in the area of evaluating the quality of synthetic instruction data.

**Strengths:**

1.The paper takes an innovative perspective by using the model’s own hidden states as a signal of data quality, removing dependence on external evaluation models or manually crafted metrics. It would be interesting to see future work analyzing how these hidden-state signals correlate with human evaluation metrics to improve interpretability.

2.Experimental results demonstrate that the proposed approach can achieve better performance with less than 10% of the data compared to full-dataset training, which substantially reduces training costs.

3.The linear separability and layer-wise comparison analyses nicely support the claim that hidden states indeed encode discriminative semantic features of high- vs. low-quality data — this is a very interesting finding.

4.The appendix includes additional experiments such as full-parameter fine-tuning, iterative filtering, inter-layer comparison, and cross-model preference studies, all of which strengthen the robustness of the method. I look forward to the authors releasing the project code after acceptance so that I can further explore this work in detail.

**Weaknesses:**

1.The method is relatively complex, as it requires extracting multi-layer hidden states and training an external CNN model, which demands significant GPU memory and computation.

2.The cross-model experiment shows that using “high-quality data” selected by Qwen2.5–7B-Instruct to fine-tune LLaMA2–7B–Chat actually led to degraded performance. This indicates that each LLM has its own internal definition of “high-quality data,” implying that CPQS must be trained separately for each target model.

**Questions:**

1.It remains unclear whether CPQS can generalize well across different model architectures or parameter scales. Can the method be effectively applied to other open-source LLMs?

2.Although the results show linear separability between high- and low-quality samples, the underlying mechanism of why hidden states reflect data quality is still largely empirical. I encourage the authors to explore this aspect more theoretically in future work.

3.Since hidden layers contribute differently to performance (see Appendix A.4.1), it would be valuable to develop an automatic strategy for determining the optimal layer combination for CPQS extraction.

---

> ### Author Response · Authors · 2025-11-21
> **Response to Reviewer eg3w (1/2)**
>
> 1.The method is relatively complex, as it requires extracting multi-layer hidden states and training an external CNN model, which demands significant GPU memory and computation.
>
> A:Thank you for your valuable comment. During training, our method processes hidden states entry by entry, releasing them immediately after use. This design minimizes memory usage during both training and inference, enhancing efficiency. In our experiments, each CNN model trains in about 30 minutes, demonstrating that our approach does not require significant GPU memory or computational resources.
>
> For a direct comparison of GPU memory usage and computational efficiency, all experiments were conducted on an NVIDIA RTX PRO 6000 GPU. The GPU memory consumption and throughput (samples per second) are summarized below, with throughput calculated over 146,224 samples based on total processing time. Our method achieves higher processing speed than both MoDs and Alpagasus, while using less GPU memory than MoDs, offering a balanced and efficient approach in terms of both speed and resource usage.
>
> | Method| GPU Memory | Throughput (samples/s) |
> |--|--|---|
> | Self| 18 GB| 4.78|
> | MoDs| 24 GB| 1.40|
> | Alpagasus| 0 GB| 1.45|
> | Superfliting | 2 GB| 34.82|
>
> 2.The cross-model experiment shows that using “high-quality data” selected by Qwen2.5–7B-Instruct to fine-tune LLaMA2–7B–Chat actually led to degraded performance. This indicates that each LLM has its own internal definition of “high-quality data,” implying that CPQS must be trained separately for each target model.
>
> A:Thank you for your thoughtful comment. We agree that this is indeed a potential issue. To address it, we recommend training a separate CNN model for each LLM. The main advantage of this approach is its extremely low cost in terms of both GPU memory and training time, as shown in the response to the previous question. For example, training a CNN for a 7B-scale model can be completed on a single RTX 4090 GPU in just about 30 minutes, with memory usage comparable to LoRA fine-tuning.
> To further validate this, we conducted additional cross-model experiments using BitCPM4-1B and Qwen2.5-32B-Instruct. For each model, we trained its own CNN to predict and filter the Reasoning-DeepSeek dataset, selecting 10k samples for fine-tuning. After that, we swapped the selected datasets between the two models and retrained them.
> The results, shown in the table below, reflect the performance of each model on four datasets, with the average score serving as the final evaluation metric. Higher values indicate better performance. As the results demonstrate, each model performs better when using data selected by its own CNN, while using cross-model data leads to reduced performance.
>
> This confirms that different LLMs have distinct internal definitions of “high-quality data”, which makes it both necessary and efficient to train a dedicated CNN selector for each target model.:
>
> |Size/ Model     | GSM8K | Math_500 | HuanEval |  GPQA | Average |
> | ----- | :---: | :------: | :------: | :---: | :--: |
> | 10k (self, 1B)  | 38.06 |   33.00  |   54.88  | 28.79 |  **38.68**  |
> | 10k (32B → 1B)  | 42.00 |   32.80  |   48.78  | 24.24 |  36.96  |
> | 10k (self, 32B) | 85.52 |   81.40  |   56.10  | 46.46 |  **67.36**  |
> | 10k (1B → 32B)  | 85.60 |   80.40  |   51.22  | 42.93 |  65.05  |

---

> ### Author Response · Authors · 2025-11-21
> **Response to Reviewer eg3w (2/2)**
>
> 3.It remains unclear whether CPQS can generalize well across different model architectures or parameter scales. Can the method be effectively applied to other open-source LLMs?
>
> A:Thank you for this valuable comment. To further assess the generalization ability of CPQS across different model architectures and parameter scales, we extended our experiments beyond the three LLMs already reported in the paper.
> To cover a broader range of model sizes, we selected one smaller-scale and one larger-scale model to complement our previous settings. Specifically, we additionally applied CPQS to BitCPM4-1B and Qwen2.5-32B-Instruct, using it to filter the Reasoning-DeepSeek dataset for high-quality data selection.
> We filtered data subsets of varying sizes (10k, 20k, etc.) using different methods and trained models on both architectures. The experimental results are shown in the table below, which follows a layout similar to Table 2 in the main text. For each trained model, we evaluated performance across four datasets and calculated the average score as the final metric, where higher values indicate better performance.
> The results demonstrate that our method consistently outperforms all baseline approaches across both small-scale (1B) and large-scale (32B) models. The detailed results are presented below.
>
> ### **BitCPM4-1B Results**
>
> | Size      | Model           | GSM8K   | Math_500   | HumanEval   | GPQA    | Average   |
> | -- | -------- | :------: | :-: | :--: | :-----: | :-------: |
> | 10k       | Self            |   38.06 |    33.00   |    54.88    |   28.79 |  **38.68**|
> | 10k       | Superfiltering   |   39.35 |    30.20   |    52.44    |   24.24 |   36.56   |
> | 20k       | Self            |   37.00 |    32.00   |    53.66    |   29.80 |  **38.12**|
> | 20k       | Superfiltering   |   36.92 |    30.09   |    54.27    |   27.27 |   37.14   |
> | 113k      | Alpagasus       |   37.38 |    30.80   |    53.66    |   25.76 |   36.90   |
>
> ### **Qwen2.5-32B-Instruct Results**
>
> | Size      | Model           | GSM8K   | Math_500   | HumanEval   | GPQA    | Average   |
> | -- | -------- | :------: | :-: | :--: | :-----: | :-------: |
> | 10k       | Self            |   85.52 |    81.40   |    56.10    |   46.46 |  **67.36**|
> | 10k       | Superfiltering   |   85.22 |    81.80   |    56.71    |   43.43 |  66.79|
> | 20k       | Self            |   84.76 |    81.60   |    53.66    |   46.46 |  **66.62**|
> | 20k       | Superfiltering   |   84.99 |    81.40   |    49.39    |   42.93 |  64.68|
> | 113k      | Alpagasus       |   84.91 |    81.00   |    53.66    |   39.90 |  64.87|
>
> 4.Although the results show linear separability between high- and low-quality samples, the underlying mechanism of why hidden states reflect data quality is still largely empirical. I encourage the authors to explore this aspect more theoretically in future work.
>
>
> A:Thank you very much for your insightful suggestion. We fully agree that the underlying mechanism of why hidden states reflect data quality deserves deeper theoretical exploration, and we plan to further investigate this direction in future work.
>
> As a preliminary step, we conducted an additional analysis to examine whether hidden states can effectively capture differences between tasks. Specifically, we selected four representative task types — math reasoning, open-domain QA, coding, and translation — and sampled 200 instances per task. We then computed both intra-task and inter-task similarities based on hidden-state representations, as shown below:
>
> | Task           | Intra-task Similarity | Inter-task Similarity | Separability (Δ) |
> | ------- | :-----: | :-----: | :-------: |
> | Math Reasoning |  0.9314|   0.7710  |0.1604 |
> | Open QA  |  0.9045  | 0.7172  |      0.1874|
> | Coding  |  0.8769| 0.7704        |      0.1065 |
> | Translation    |  0.8359  |         0.7281    |      0.1077      |
>
> As the results indicate, the intra-task similarities are consistently higher than inter-task similarities, suggesting that hidden states encode clear task-specific distinctions. This observation supports the idea that hidden representations inherently capture meaningful structure about data and task quality, providing an empirical foundation for future theoretical analysis.
>
> 5.Since hidden layers contribute differently to performance (see Appendix A.4.1), it would be valuable to develop an automatic strategy for determining the optimal layer combination for CPQS extraction.
> Thank you for your thoughtful suggestion. We completely agree that different hidden layers contribute unequally to model performance, and that developing an automatic strategy for selecting the optimal layer combination would be highly valuable.
>
> A:In future work, we plan to explore adaptive or learnable layer-weighting mechanisms to automatically determine which layers provide the most informative hidden representations for CPQS extraction. We believe this direction could further enhance both the robustness and generalization of our method.

---

### Public Comment · ~Zihao_Chen10 · 2025-11-14

Thank you for sharing this fascinating paper—it has been quite enlightening. I've recently been exploring data filtering for fine-tuning large models. If you have some spare time, would you be willing to address my questions about this paper?

Q1: I've read many papers on instruction data filtering, and they consider not only data quality but also factors like diversity. This paper doesn't seem to mention diversity—is it because diversity has a relatively minor impact on model performance?

Q2: The experimental section shows that Llama-3.2-1B-Ins and Qwen-2.5-1.5B-Ins to generate low-quality samples. However, according to their respective official technical reports, these models outperform the Llama-2-7b used in the experiments. Would the low-quality data generated by the aforementioned two models also be classified as low-quality data by Llama-2-7b?

I hope my questions aren't bothering you. Thank you for your time!

---

### Author Response · Authors · 2025-12-03
**Global Response to the Reviewers and ACs**

We sincerely thank the reviewers for their time and valuable feedback, which has significantly improved our work. We appreciate the recognition of our approach's innovative aspects, especially the use of the model’s own hidden states to evaluate data quality, which removes the need for external evaluation models or manually crafted metrics (Reviewers eg3w, ZLxp, vfku). We are also encouraged by the acknowledgment of our method’s empirical efficiency, achieving comparable performance with less than 10% of the data, which is especially useful in resource-constrained scenarios (Reviewers eg3w, ZLxp). Additionally, the reviewers highlighted the effectiveness of our experimental validation, including the layer-wise comparison and appendix experiments, which strengthen our method’s robustness (Reviewers eg3w, ZLxp). We are grateful for the recognition of the practicality and timeliness of our work, particularly in reducing compute and data costs for fine-tuning large language models (Reviewers ZLxp, eg3w). Finally, we appreciate the positive comments on the clarity and contribution of our paper (Reviewers vfku, jCwj).

We have provided detailed, point-by-point responses to the reviewers; however, here is a summary of the principal points addressed in the reviews.

### Reviewer eg3w

* **Complexity and GPU memory usage**: We have clarified how our method efficiently manages memory and computational resources (Response to Comment 1).
* **Cross-model experiments and model-specific definitions of "high-quality data"**: We provided experimental evidence supporting the need for separate CNN models for each LLM (Response to Comment 2).
* **Generalization across different architectures**: We extended experiments to show the robustness of our method (CPQS) across various model architectures and parameter scales (Response to Comment 3).
* **Theoretical exploration of hidden states reflecting data quality**: We provided a preliminary analysis and outlined future plans for further investigation (Response to Comment 4).
* **Automatic strategy for selecting the optimal layer combination**: We outlined future work to explore this direction (Response to Comment 5).

### Reviewer vfku

* **In-depth analysis of method’s improvement over baselines**: We clarified that Alpagasus was used only as a reference and explained how our method outperforms it (Response to Comment 1).
* **Justification of CNN structure**: We conducted experiments comparing CNN, MLP, and Transformer models, showing CNN's superior performance (Response to Comment 2).
* **Models of different sizes**: We extended experiments to include various model sizes and showed that each model performs better with its own CNN (Response to Comment 3).

### Reviewer jCwj

* **Computational cost analysis**: We conducted experiments to compare GPU memory usage and throughput across methods, demonstrating our method’s efficiency (Response to Comment 1).
* **Comprehensive comparative study**:  We expanded experiments to include additional baselines such as DS2, Deita, and SelectIT, highlighting our method’s competitive advantage (Response to Comment 2).
* **Novelty of the motivation**: We clarified that our method’s novelty lies in using the model’s own hidden states for data quality assessment (Response to Comment 3).
* **Shared synthetic "quality" dataset**: We explained that "quality" is model-specific and varies across LLMs (Response to Comment 4).

### Reviewer ZLxp

* **Synthetic data for training the classifier**: We clarified that our approach follows prior work and demonstrated strong alignment with human perceptions of instruction quality (Response to Comment 1).
* **Limited scope of evaluation**: We extended experiments to include different model sizes and demonstrated consistent performance (Response to Comment 2).
* **Cost and efficiency claims**: We provided a detailed comparison of GPU memory usage and computational efficiency (Response to Comment 3).
* **Potential bias in data selection**: We conducted a diversity analysis and ensured broad task/domain coverage (Response to Comment 4).
* **Robustness to architecture shifts**: We demonstrated that CPQS performs well across various architectures and compared classifier models (Response to Comment 7).
* **Detailed compute cost comparisons**: We provided a breakdown of GPU usage and costs across different stages (Response to Comment 8).
* **Analysis of task coverage and diversity**: We conducted a hidden-state similarity analysis to confirm diverse task coverage (Response to Comment 9).
* **Per-benchmark breakdowns for downstream tasks**: We provided detailed deployment and experiment specifications. (Response to Comment 10).

We believe that these revisions address all of the reviewers' concerns and significantly strengthen our work. Thank the reviewers again for their valuable feedback!

---

### Meta-Review · Area_Chair_Grci · 2026-01-05

**Summary:**

This paper introduces CPQS-Tuning, a data filtering algorithm that leverages a target model's hidden states to identify high-quality instruction data. By training a classifier on these internal representations, the method selects samples that align with the model's own perception, removing the need for external evaluation models. Empirical results confirm that the method significantly enhances data efficiency, consistently outperforming state-of-the-art selection techniques while achieving competitive performance using only a small fraction of the total training data.

**Reviewer Concerns:**

Addressed by Rebuttal:
- The authors provided a cost analysis demonstrating that their method achieves higher throughput than previous frameworks while maintaining a manageable GPU memory footprint.
- Authors extended experiments to include models ranging from 1B to 32B parameters, proving the method's robustness across different scales.
- The authors incorporated additional modern baselines, with CPQS-Tuning consistently showing superior average performance.
- The use of a CNN over other architectures was justified by its superior ability to extract local structural patterns from hidden states.

Outstanding Concerns:

- One reviewer noted that the classifier still utilizes synthetic data tiers. While the authors provided human evaluation showing high alignment with these tiers, exploring purely human-annotated utility remains a direction for future work.

**Reviewer Scores:**

Reviewer eg3w: Maintained a 10, praising the innovative use of hidden states for quality evaluation and the compelling data efficiency results.

Reviewer vfku: Maintained Marginally Above after participating in the discussion and confirming that concerns regarding architecture and model scale were resolved.

Reviewer ZLxp: Maintained Marginally Above; the rebuttal provided human-validation data and empirical evidence of success on 32B models to mitigate concerns regarding potential bias.

Reviewer jCwj: Maintained a 2 (Reject); the authors addressed the initial technical criticisms by providing missing baseline comparisons and a detailed computational cost analysis.

---

### Decision · Program_Chairs · 2026-01-26

Accept (Poster)